# On the Intrinsic Limits of Transformer Image Embeddings in Non-Solvable Spatial Reasoning

**Siyi Lyu** [1]   **Quan Liu** [1]   **Feng Yan** [1]

## Abstract

Vision Transformers (ViTs) excel in semantic recognition but exhibit systematic failures in spatial reasoning tasks such as mental rotation. While often attributed to data scale, this work argues that the limitation arises from the intrinsic circuit complexity of the architecture. By formalizing spatial understanding as a **Group Homomorphism Problem**—where latent embeddings preserve the algebraic structure of physical transformations acting on images—we identify a fundamental computational bottleneck. Specifically, for non-solvable groups (e.g., $SO(3)$), maintaining such structure-preserving embeddings is lower-bounded by the Word Problem, which is $NC^1$-complete. In contrast, constant-depth ViTs with polynomial precision are strictly bounded by the complexity class $TC^0$. Under the standard conjecture $TC^0 \subsetneq NC^1$, a **complexity boundary** emerges: constant-depth architectures lack the logical depth required to capture non-solvable spatial structures in a single forward pass. To empirically validate this theoretical gap, we propose the **Latent Space Algebra (LSA)** benchmark, which reveals a significant degradation in ViT representations as the compositional depth of non-solvable tasks increases.

## 1. Introduction

Vision Transformers (ViTs) have fundamentally reshaped computer vision (Vaswani et al., 2017; Carion et al., 2020; Dosovitskiy et al., 2021; Khan et al., 2022). While these architectures have achieved state-of-the-art performance in semantic tasks, persistent failures in spatial reasoning have been observed (Stogiannidis et al., 2025; Khemlani

et al., 2025; Chen et al., 2025b). Recent benchmarks indicate that even massive foundation models struggle with geometric transformations—such as mental rotation and relative positioning—beyond simple edge cases (Keremis et al., 2025; Kong et al., 2025). This raises a critical question: *Is this failure a result of insufficient data or an intrinsic complexity barrier?*

This work identifies the latter, proposing that standard ViT embeddings are theoretically constrained by their constant circuit depth, which limits their capacity to model the algebraic structure of complex spatial transformations.

To formalize this, spatial understanding is defined as the acquisition of a **Group Homomorphism**. A robust, object-agnostic embedding should ensure that the relationships between latent representations faithfully preserve the composition law of the underlying geometric group $G$ (e.g., $SO(3)$). Under this framework, for a neural network to maintain such a structure-preserving space, it must implicitly solve the Word Problem for that group.

Leveraging Circuit Complexity Theory, a critical bottleneck is identified. By Barrington's Theorem, the Word Problem for finite non-solvable groups (such as the icosahedral subgroup of $SO(3)$) is $NC^1$-complete, requiring logarithmic logical depth to resolve serial dependencies. In contrast, standard ViT encoders, operating with constant depth and polynomial precision, are strictly bounded by the complexity class $TC^0$. Under the standard conjecture $TC^0 \subsetneq NC^1$, a **complexity boundary** emerges: constant-depth architectures lack the requisite logical depth to capture non-solvable spatial structures. This implies that ViTs rely on shallow approximations rather than mastering the underlying group isomorphism.

These theoretical limits are empirically validated via the **Latent Space Algebra (LSA)** benchmark. By employing a recursive linear probing protocol, it is demonstrated that while ViT representations maintain fidelity for abelian transformations, they undergo a structural degradation on non-solvable relations as the compositional depth increases.

**Contributions.** This study provides three primary contributions: **(1) Algebraic Formalization:** Spatial representation is cast as a group homomorphism problem, categorizing task

[1]School of Electronic Science and Engineering, Nanjing University, Nanjing, China. Correspondence to: Feng Yan <fyan@nju.edu.cn>.

*Proceedings of the $43^{rd}$ International Conference on Machine Learning*, Seoul, South Korea. PMLR 306, 2026. Copyright 2026 by the author(s).

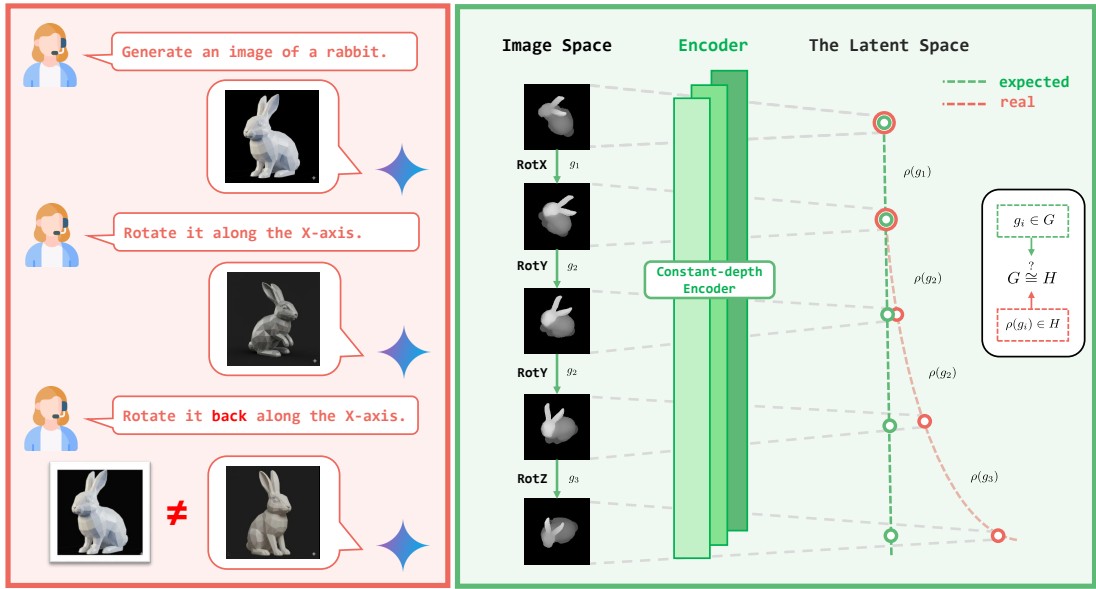

*Figure 1.* **From Empirical Spatial Failures to the Homomorphism Alignment Problem. (Left)** The Spatial Reasoning Problem: Evidence of foundation models failing simple mental rotation tasks. **(Right)** The Homomorphism Alignment Problem: Can a constant-depth encoder map observations transformed by group $G$ to a latent space where induced dynamics $H$ are isomorphic to $G$ ($H \cong G$)? The analysis demonstrates that for non-solvable groups, this isomorphism is prohibited by architectural complexity constraints.

hardness by the solvability of the underlying transformation group. **(2) Complexity Boundary Proof:** It is formally established that constant-depth ViTs are theoretically incapable of capturing non-solvable structures (e.g., 3D rotations) under standard complexity conjectures. **(3) Algebraic Benchmark and Empirical Validation:** Departing from conventional task- or phenomenon-driven paradigms, we propose the LSA benchmark to evaluate spatial reasoning through the lens of algebraic structures. Empirical evaluations on LSA successfully validate our theoretical insights, demonstrating a significant degradation in ViT representations when modeling non-solvable dynamics.

## 2. Related Work

### 2.1. Spatial Reasoning Failures in Vision Transformers

Despite their semantic success, ViTs exhibit systemic failures in spatial reasoning. Empirical studies show that large-scale Vision-Language Models (VLMs) struggle with basic prepositions, often failing to distinguish relative positioning better than chance (Subramanian et al., 2022; Lepori et al., 2024; Stogiannidis et al., 2025; Kong et al., 2025). While models may excel at absolute spatial queries, they falter on compositional benchmarks like Winoground, acting as "pattern matchers" rather than robust reasoners (Thrush et al., 2022). This deficit extends to dynamic transformations, where performance drops significantly on non-canonical

3D rotations and abstract geometric tasks (Li et al., 2025; Tuggener et al., 2025; Keremis et al., 2025; Khemlani et al., 2025; Chen & Artzi, 2025). Unlike prior work attributing these issues to optimization artifacts or attention misalignment (Qi et al., 2025; Chen et al., 2025b), this study investigates whether the architecture possesses the computational capacity to implement spatial algorithms using Circuit Complexity.

### 2.2. Computational Expressivity of Transformer Architecture

Circuit Complexity provides a rigorous framework for mapping Transformer limits. Theoretical analyses establish that standard constant-depth Transformers with polynomial precision are restricted to the DLOGTIME-uniform $\mathsf{TC}^0$ class (Hahn, 2020; Liu et al., 2023; Merrill & Sabharwal, 2023b). This bound precludes the execution of inherently serial computations or deep hierarchical reasoning within a single forward pass (Merrill & Sabharwal, 2023a; Chiang, 2025). Further studies link these constraints to failures in solving reachability or circuit evaluation tasks, limitations that persist in alternative architectures like State-Space Models (SSMs) and Mamba-like architectures (Peng et al., 2024; Merrill et al., 2024; Chen et al., 2024). However, while existing literature focuses on formal languages and logic, the application of circuit complexity to explain spatial reasoning via group-theoretic structures remains unexplored.

# 3. Preliminaries

Evaluating the spatial reasoning capabilities of modern visual encoders remains a challenge, as conventional evaluations frequently treat spatial understanding as phenomenological curve-fitting to specific tasks. To establish a more rigorous metric, this work draws on the philosophy of Henri Poincaré and the Erlangen Program (Klein, 1893; Poincaré, 1905). From this perspective, "space" is not merely a collection of static coordinates, but is intrinsically defined by the invariants under a specific group of transformations. Consequently, we posit that mastering spatial relationships is mathematically equivalent to capturing these underlying group structures. Evaluating true spatial cognition, therefore, arguably requires verifying whether a model's latent representation can preserve such invariant algebraic properties.

The operational paradigm under investigation is the standard feed-forward execution. Consider images connected by sequential group transformations (e.g., image A rotates to B, and then to C). Mainstream encoders map each image to its respective latent representation via a single forward pass. A faithful spatial embedding requires these latents to satisfy the corresponding algebraic constraints dictated by the physical transformations. Our framework investigates the intrinsic limit of this paradigm: can a single forward pass truly embed images into a space governed by such algebraic constraints?

To systematically answer this, this section establishes the mathematical foundations by introducing three interlocking concepts. First, we analyze the **Solvability Hierarchy** of groups (Section 3.1), which characterizes the degree of algebraic entanglement in physical transformations and reveals that non-commutative operations induce serial dependencies. Second, we introduce the **Word Problem** (Section 3.2), a formal algebraic paradigm that encapsulates the challenge of resolving these cumulative serial dependencies. Third, we utilize **Circuit Complexity** (Section 3.3) to measure the computational hardness of the Word Problem, bounded by Barrington's Theorem, making the algebraic difficulty directly comparable to the expressivity limits of neural architectures.

## 3.1. Task Difficulty: The Solvability Hierarchy

To investigate whether latent representations can effectively preserve algebraic structures, it is crucial to recognize that different transformation groups exhibit distinct mathematical properties, with commutativity being of primary importance. From an algebraic perspective, this commutativity structure heavily influences the dependency between sequential operations.

Intuitively, if operations commute, their order is irrele-

vant, allowing for parallel aggregation; if they do not, the operations are algebraically entangled, yielding strict serial dependencies. This structural property is formalized by the Derived Series of the group $G$. The series tracks the depth of non-commutativity using the commutator $[g, h] = g^{-1}h^{-1}gh$, which quantifies the difference between executing $g$ then $h$ versus $h$ then $g$.

Formally, for any subgroup $H \subseteq G$, the derived subgroup $[H, H]$ is defined as the subgroup generated by the set of all commutators of elements in $H$, i.e., $[H, H] = \langle\{[g, h] \mid g, h \in H\}\rangle$. The Derived Series is then defined recursively as $G^{(0)} = G$ and $G^{(k+1)} = [G^{(k)}, G^{(k)}]$. Termination of the Derived Series at the trivial subgroup indicates that all higher-order non-commutative dependencies can be resolved into simpler components, giving rise to the following classification (Hall, 2015):

**Level 1: Abelian – e.g., 2D Translation.** Operations commute ($gh = hg$), so the commutator is trivial ($[g, h] = e$). The derived series terminates immediately ($G^{(1)} = \{e\}$).

**Level 2: Solvable Non-Abelian – e.g., 2D Rigid Motion.** Order matters ($gh \neq hg$), but the transformation can be decomposed into a finite hierarchy of abelian steps. Specifically, the group of rigid transformations ($\mathbb{R}^2 \rtimes SO(2)$) has a derived series that terminates at the identity in finite steps ($G^{(k)} = \{e\}$), allowing for relatively shallow recursive evaluation.

**Level 3: Non-Solvable – e.g., 3D Rotation.** Operations are deeply entangled and cannot be decoupled into abelian layers. The derived series never reaches the identity ($G^{(k)} \neq \{e\}$), instead stabilizing at a complex subgroup. Crucially for vision, the continuous group $SO(3)$ contains discrete subgroups isomorphic to the non-solvable group $A_5$. This algebraic "deadlock" prevents decomposition, imposing a fundamental barrier to parallelization.

## 3.2. Formalizing the Dependency: The Word Problem

The strict serial dependency generated by non-commutative spatial operations is mathematically formalized by the **Word Problem**.

**Definition 3.1** (The Finite Group Word Problem)**.** Let $G$ be a finite group (a group containing a finite number of elements) generated by a set $\Sigma$. The Word Problem over $G$ is the decision problem of determining, for an input sequence of generators $w = (g_1, \ldots, g_n) \in \Sigma^n$, whether their product evaluates to the identity element:

$$g_n \ldots g_2 g_1 \overset{?}{=} e \qquad (1)$$

While spatial reasoning typically involves evaluating a specific final physical state (e.g., $z_{\text{final}} = (g_N \ldots g_2 g_1) \cdot z_{\text{init}}$), for finite groups, this evaluation is computationally equiva-

lent to the decision-based Word Problem (Barrington, 1986; Beaudry et al., 1997). Crucially, if an encoder's latent representation implicitly contains the information required to solve the Word Problem, it indicates that the representation has successfully encoded the underlying group structure. By mapping the spatial task to the Word Problem, the physical challenge is translated into a standardized algebraic paradigm.

### 3.3. Measuring Capability: Circuit Complexity

While the Word Problem formalizes the serial dependency, **Circuit Complexity** provides the theoretical framework to measure the computational hardness of executing such operations and to bound the expressivity of neural architectures. It is essential to emphasize that in this context, circuit depth refers to an asymptotic scaling relative to the input sequence length $N$, rather than a fixed structural layer count. For instance, a deployed ResNet-152, regardless of how physically deep it is, merely executes a static computational graph and possesses a strictly constant circuit depth with respect to $N$.

Two complexity classes are central to matching parallel versus sequential capabilities:

**The Transformer Limit** ($\mathsf{TC}^0$). This class comprises problems solvable by constant-depth, polynomial-size threshold circuits. This class fundamentally characterizes the capability for highly parallel computations. Standard Transformer encoders with a fixed number of layers and polynomial precision are strictly contained within $\mathsf{TC}^0$ (Merrill & Sabharwal, 2023b; Chiang, 2025). In the framework of circuit complexity, precision refers to the number of bits required to represent intermediate numerical values. Practical neural network implementations utilizing fixed bit-width formats, such as standard 32-bit floating-point (`float32`), operate with strictly constant precision ($O(1)$ bits relative to the sequence length $N$), which trivially satisfies this polynomial bound. This places a hard ceiling on their capacity to execute serial algorithms, modeling the capability of a single forward pass that executes a static computational graph.

**The Recursive Depth Requirement** ($\mathsf{NC}^1$). This class allows for a logical depth of $O(\log N)$, where $N$ is the input size. It characterizes computations that demand deep hierarchical or sequential dependencies that scale with the input.

**The Relationship and Conjecture** ($\mathsf{TC}^0 \subsetneq \mathsf{NC}^1$). By definition, $\mathsf{TC}^0 \subseteq \mathsf{NC}^1$, meaning any problem solvable by shallow parallel circuits can be simulated by deeper sequential circuits. However, it is a standard and widely accepted conjecture in complexity theory that this inclusion is strict ($\mathsf{TC}^0 \subsetneq \mathsf{NC}^1$). If this conjecture were false (i.e., $\mathsf{TC}^0 = \mathsf{NC}^1$), it would imply that deep sequential operations

could be perfectly parallelized into shallow, constant-depth circuits without an exponential blow-up in circuit size. In the context of neural architectures, this would effectively collapse the boundary between bounded feed-forward networks and deep recurrent processes, implying that a single forward pass could simulate arbitrary loops of recursive reasoning. Relying on this separation conjecture mathematically formalizes the physical intuition that strict serial dependencies cannot be fully bypassed by parallel layers.

**Barrington's Theorem** (Barrington, 1986) directly places the evaluation of the Word Problem into the metric space of Circuit Complexity, thereby bringing the analysis of algebraic structure comprehension firmly into the realm of computational complexity theory. The theorem establishes that for any non-solvable finite group (such as $A_5$), solving the Word Problem is $\mathsf{NC}^1$-complete. This definitively indicates that the computational complexity of the task intrinsically scales with the sequence length $N$ of the generators, formalizing the exact bottleneck required to evaluate the final transformed state.

## 4. Theoretical Analysis

This section formally analyzes the computational complexity required to learn a generalized spatial embedding.

**Theoretical Roadmap.** The proof proceeds in three logical steps: First, we formalize spatial understanding as a Group Homomorphism (Section 4.1). Second, we demonstrate that if an encoder successfully satisfies this homomorphism in a single forward pass, its output latent representations must implicitly contain the algebraic information necessary to solve the Word Problem (Section 4.2). Finally, by bridging the theoretical lower bound of the Word Problem ($\mathsf{NC}^1$) with the architectural upper bound of static feed-forward networks ($\mathsf{TC}^0$), we establish a fundamental complexity barrier, preventing constant-depth architectures from fully capturing non-solvable spatial structures (Section 4.3).

### 4.1. Formalizing Spatial Understanding

Spatial understanding is defined here as the capability to model the generative mechanism of geometric transformations. Let $\mathcal{X} \subseteq \mathbb{R}^{d_{\mathrm{img}}}$ denote the observation space and $G$ be a transformation group acting on $\mathcal{X}$. To ensure compositionality and generalization, the visual embedding is modeled as a **Group Homomorphism**.

**Definition 4.1** (Homomorphic Spatial Embedding). Let $g$ represent the composite transformation resulting from a sequence of $N$ generators $(g_1, \ldots, g_N)$, i.e., $g = g_N \ldots g_1$. An encoder $E : \mathcal{X} \to \mathbb{R}^d$ computes a Homomorphic Spatial Embedding if there exists a faithful representation $\rho : G \to \mathrm{GL}(d, \mathbb{R})$ such that:

$$E(g \cdot I) = \rho(g)E(I), \quad \forall g \in G, I \in \mathcal{X} \qquad (2)$$

Here, $\mathrm{GL}(d, \mathbb{R})$ denotes the General Linear Group, consisting of all $d \times d$ invertible matrices. The specific requirement for the target space to be $\mathrm{GL}(d, \mathbb{R})$ ensures that the learned latent operations are algebraically reversible and capable of modeling the full range of linear compositions, preserving the group structure of the original transformations.

Satisfying this homomorphism serves as a necessary condition for spatial cognition. Given that space itself is an invariant quantity independent of the environment, preserving its algebraic structure is essential for generalizable spatial reasoning. This constraint forces the encoder to capture the generative group law, which is the structural requirement to handle the combinatorial explosion of composite states where memorization inevitably fails.

### 4.2. Computational Complexity Bounds

The following analysis establishes the opposing complexity bounds for the architecture and the task.

**Architectural Upper Bound.** The expressivity of the ViT architecture is characterized using Circuit Complexity.

**Lemma 4.2** (ViT Circuit Complexity under Polynomial Precision). *A standard Vision Transformer encoder, operating with constant depth $L$ and polynomial precision, lies strictly within the complexity class* DLOGTIME-*uniform* $\mathsf{TC}^0$.

*Proof.* The ViT architecture consists of two stages: Input Projection and Transformer blocks. The Input Projection maps pixel patches to vectors via linear operations which involve only the addition and multiplication of rational numbers. Such arithmetic operations can be strictly simulated by $\mathsf{TC}^0$ circuits without approximation error (Chiang, 2025). Subsequently, the Transformer Blocks consist of Self-Attention and MLPs. Recent theoretical analyses by Merrill & Sabharwal (2023b) and Chiang (2025) have established that these blocks, when operating with precision of $O(\mathrm{poly}(n))$, can be simulated by DLOGTIME-uniform $\mathsf{TC}^0$ circuits, where $n$ is the input size. As standard implementations utilize a constant number of bits (e.g., `float32` or `bf16`), which is a subset of the allowed polynomial precision, and since $\mathsf{TC}^0$ is closed under composition, the entire ViT pipeline remains strictly within DLOGTIME-uniform $\mathsf{TC}^0$. □

**Task Lower Bound.** The difficulty of the embedding task is determined by the algebraic structure of $G$. Crucially, the following lemma evaluates the informational requirement of a single forward pass.

**Lemma 4.3** (Reduction to the Word Problem). *Let $G$ be a finite group acting on the image space. If an encoder $E$ satisfies Definition 4.1 (Homomorphic Spatial Embedding), then the computational complexity of the task is lower-bounded by the Word Problem for $G$.*

*Proof.* Consider a sequence of generators $(g_1, \ldots, g_N)$ acting on an initial reference image $I_0$. Here, $N$ denotes the compositional depth and should not be confused with the resolution of the image. Let $z_0 = E(I_0)$ be the known embedding of the initial state. The final image after the sequence of transformations is given by $I_N = (g_N \ldots g_1) \cdot I_0$. If the encoder $E$ successfully completes the homomorphic embedding task, it produces a final embedding $z_N = E(I_N)$. According to Definition 4.1, this output must satisfy:

$$z_N = \rho(g_N \ldots g_1)z_0 = (\rho(g_N) \ldots \rho(g_1))z_0$$

Since $z_0$ is known and fixed, and the encoder only observes the final physical state $I_N$, correctly computing $z_N$ for an arbitrary sequence of length $N$ necessitates that the single forward pass intrinsically embeds the outcome of the iterated matrix product $\rho(g_N) \ldots \rho(g_1)$. Crucially, the definition requires $\rho$ to be a faithful representation, implying it is an injective homomorphism that preserves the group structure isomorphically (i.e., $\rho(g) = I \iff g = e$). This isomorphism guarantees that the computational complexity of evaluating the iterated product in the matrix group $\rho(G)$ is equivalent to evaluating the product sequence in the abstract group $G$. Furthermore, the evaluation problem over finite groups is computationally equivalent to the Word Problem (Barrington, 1986; Beaudry et al., 1997). Consequently, the ability of the encoder to map any transformed image $I_N$ to its correct latent algebraic coordinate $z_N$ in a single forward pass implies that its computation inherently contains the complete information required to solve the Word Problem, thereby inheriting its computational hardness. □

*Remark* 4.4 (From Discrete Subgroups to Continuous Groups). Although Lemma 4.3 leverages finite non-solvable groups (e.g., $A_5$), computer vision tasks typically involve continuous groups such as $\mathrm{SO}(3)$. The lower bound remains valid via subgroup restriction. If a network can compute a homomorphic embedding for the entire continuous group $G$, it must necessarily succeed for any subset of inputs drawn from a subgroup $H \subset G$. Since $\mathrm{SO}(3)$ contains discrete subgroups isomorphic to $A_5$, and the Word Problem for $A_5$ is $\mathsf{NC}^1$-complete, the general task for $\mathrm{SO}(3)$ inherits this $\mathsf{NC}^1$-hardness.

### 4.3. The Complexity Barrier

Combining the architectural limit ($\mathsf{TC}^0$) with the task lower bound ($\mathsf{NC}^1$), the main theoretical result follows.

**Theorem 4.5** (The Non-Solvable Barrier). *Let $G$ be a transformation group containing a finite non-solvable subgroup where the Word Problem is $\mathsf{NC}^1$-complete. Under the standard complexity conjecture that $\mathsf{TC}^0 \subsetneq \mathsf{NC}^1$, a constant-depth Vision Transformer with polynomial precision cannot implement a Homomorphic Spatial Embedding for $G$ in a single forward pass.*

*Proof.* The proof proceeds by contradiction. Assume there exists a ViT encoder $f \in \mathcal{F}_{\text{ViT}}$ that implements a Homomorphic Spatial Embedding for a non-solvable group $G$. By Lemma 4.3, such an encoder effectively solves the Iterated Group Product problem for any sequence of generators in $G$. Since $G$ is non-solvable, it contains a finite subgroup for which the Word Problem is $\mathsf{NC}^1$-complete (Barrington, 1986). This implies that $f$ must be able to simulate any problem in $\mathsf{NC}^1$. However, by Lemma 4.2, any function computed by a standard fixed-depth ViT lies strictly within the complexity class $\mathsf{TC}^0$. Consequently, the existence of such an embedding would imply $\mathsf{NC}^1 \subseteq \mathsf{TC}^0$. This contradicts the standard separation conjecture $\mathsf{TC}^0 \subsetneq \mathsf{NC}^1$. Therefore, the initial assumption must be false: a standard ViT, operating via a single forward pass, lacks the intrinsic logical depth to capture the algebraic structure of non-solvable groups. $\square$

## 4.4. Boundary Analysis

Our result establishes a hard barrier for standard constant-depth architectures. Here, we analyze three common architectural extensions, demonstrating that such structural augmentations do not suffice to break the barrier.

### 4.4.1. CHAIN-OF-THOUGHT

While standard Vision Transformers operate as purely feed-forward, constant-depth encoders, a potential counter-argument suggests that Chain-of-Thought (CoT) unrolling could bypass the $\mathsf{TC}^0$ limitation by expanding the effective circuit depth to $O(N)$, theoretically enabling $\mathsf{NC}^1$ simulation (Merrill et al., 2024). However, this extension effectively reverts the architecture to a deep recurrent model, resurrecting the historical challenges of RNNs.

Unlike discrete symbolic reasoning, spatial embedding operates on continuous manifolds where approximation errors compound exponentially with depth ($\epsilon_{\text{total}} \sim (1 + \epsilon)^N$). Consequently, while CoT theoretically raises the complexity ceiling, it forces the model to confront the very issues—analog drift and optimization instability over long horizons—that non-recurrent architectures were designed to eliminate. Thus, the barrier remains practical: one must choose between the insufficient expressivity of constant depth or the optimization pathology of deep recurrence.

### 4.4.2. POSITIONAL ENCODINGS

Positional Encodings (PE) inject sequence order but act strictly as pre-processing steps that do not increase the logical depth of the circuit. **Absolute PE** ($x_i' = x_i + p_i$) involves the element-wise addition of data-independent vectors, which is an $\mathsf{AC}^0$ operation (Bergsträßer et al., 2024). **Rotary PE (RoPE)** applies a rotation $R_{\theta,i}$. Crucially, for a fixed position $i$, $R_{\theta,i}$ is a constant matrix, and multiplying

variable vectors by constant matrices is an $\mathsf{TC}^0$ map (Chen et al., 2025a). Since $\mathsf{TC}^0$ is closed under composition with constant-depth circuits, a ViT equipped with PE acts as a pre-computed lookup table and remains strictly within $\mathsf{TC}^0$.

### 4.4.3. SE(3)-EQUIVARIANT NETWORKS AND INDUCTIVE BIAS

Specialized architectures like SE(3)-Transformers (Thomas et al., 2018; Tai et al., 2019; Fuchs et al., 2020; Romero & Cordonnier, 2021; Xu et al., 2023) explicitly bake in geometric structure via their kernel definition:

$$K_{J,J'}(\mathbf{x}) = \sum_{\ell} \underbrace{w_\ell(\|\mathbf{x}\|)}_{\substack{\text{learnable} \\ \text{radial profile}}} \cdot \left( \underbrace{Y_\ell(\hat{\mathbf{x}}) \otimes C_{\text{CG}}}_{\substack{\text{fixed} \\ \text{geometric structure}}} \right)_{J,J'} \quad (3)$$

This formulation reveals that the non-solvable algebra of SO(3) is encoded entirely in the fixed Spherical Harmonics ($Y_\ell$) and Clebsch-Gordan coefficients ($C_{\text{CG}}$). However, while these constants provide the local multiplication rule (the group table), they do not confer the compositional depth required to solve the Word Problem for long sequences. Solving the Word Problem for a sequence of length $N$ requires recursively applying these group operations, necessitating a logical depth of $\Omega(\log N)$. Regarding the learnable dynamics (the radial weights $w$ or message passing), recent theoretical work by Cao et al. (2025) proves that standard equivariant layers can be simulated by uniform threshold circuits. Consequently, such models theoretically cannot solve the Word Problem for non-solvable groups via their learnable dynamics.

## 5. Experimental Verification

To empirically validate the theoretical bounds established in Theorem 4.5, the **Latent Space Algebra (LSA)** benchmark is designed. This framework probes whether standard backbones learn structure-preserving embeddings across the solvability hierarchy, strictly testing combinatorial generalization under recursive group operations. To complement the learned approach, Orthogonal Procrustes Analysis (OPA) is employed as a deterministic control. By quantifying intrinsic linearity without training, OPA rules out optimization artifacts, verifying that recursive failures stem from fundamental structural deficits rather than probe training issues.

### 5.1. The Latent Space Algebra (LSA) Benchmark

A hierarchy of three synthetic datasets is constructed, each governed by a group $G$ corresponding to the algebraic structures defined in the Preliminaries. This progression isolates the point where circuit depth becomes a bottleneck.

**Level 1: Abelian** ($G \cong \mathbb{Z}^2$)**.** The baseline structure consists

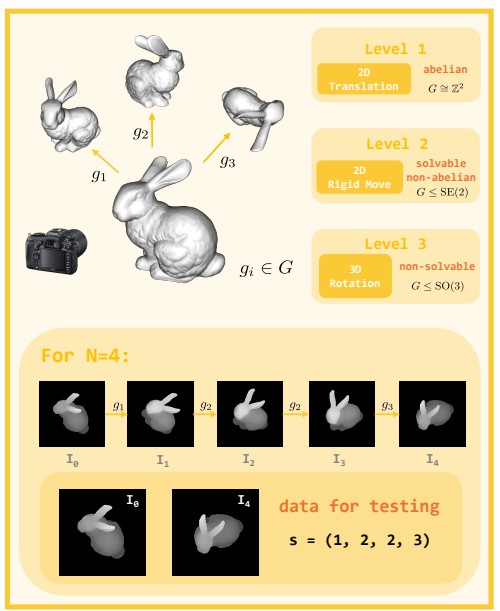 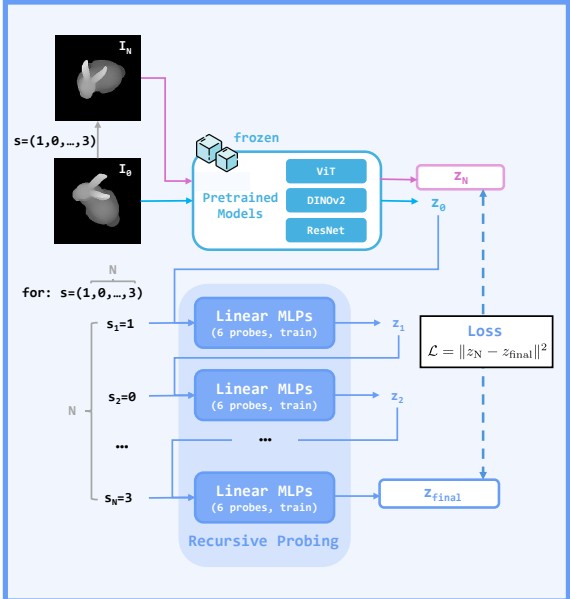

*Figure 2.* **Overview of the Recursive Linear Probing Protocol. (Left)** The Data Generation Process: Image trajectories are sampled using operators from three complexity levels, yielding tuples $(I_0, I_N, S)$ (e.g., for $N = 1$, $S = (1)$; for $N = 4$, $S = (1, 2, 2, 3)$). **(Right)** The Recursive Probing (test mode): During training ($N = 1$), the probe minimizes the error between the predicted $\hat{z}_1$ (from $z_0, s$) and the ground truth $z_1$ given by frozen pretrained model. Crucially, during testing ($N > 1$), probes are forced to recursively apply the learned transition $N$ times to $z_0$ using the sequence $S$ to generate $z_{\text{final}}$. The loss is then computed as the distance between this recursively generated embedding and the ground truth $z_N$ from the frozen backbone.

of 2D translations on a lattice. Since translations commute $(T_x T_y = T_y T_x)$, the group is abelian and solvable. Standard constant-depth circuits are theoretically capable of modeling this structure.

**Level 2: Solvable Non-Abelian ($G \leq \mathrm{SE}(2)$).** This level introduces 2D rigid motions, combining rotation and translation. While the operations do not commute, the group is structurally constrained and admits a derived series that terminates at the identity. This tests the capability to handle non-commutative operations that are strictly decomposable into finite abelian extensions.

**Level 3: Non-Solvable ($G \leq \mathrm{SO}(3)$).** The theoretical barrier involves 3D rotations around the X, Y, and Z axes. These generators produce a dense subset of $\mathrm{SO}(3)$, which contains finite non-solvable subgroups. Modeling sequences in this domain requires logical depth beyond $\mathsf{TC}^0$.

Data generation employs a unified recursive pipeline based on random walks of length $N_{\max} = 20$. To strictly ensure that training and testing explore the identical state space, we partition these trajectories by depth: the model is trained solely on atomic transitions ($N = 1$) and evaluated on recursive sequences ($N \in \{2, \ldots, 20\}$) extracted from the same generative paths. Detailed data formatting is visualized in Figure 2.

To ensure validity, two rigorous controls are enforced:

**Visual Injectivity.** While a standard 3D-to-2D projection does not naturally preserve underlying algebraic structures, our benchmark overcomes this by explicitly rendering asymmetric 3D objects (from the Stanford Scanning Repository (Curless & Levoy, 1996)). This ensures a strictly injective mapping from the abstract group state to the 2D image space, guaranteeing that the rendered images inherently retain the mathematical relationships of the external 3D group. Consequently, the visual reasoning task rigorously inherits the computational hardness associated with evaluating the underlying combinatorial product (the formal proof of this structure preservation is detailed in Appendix A.1).

**Combinatorial Generalization.** By construction, the recursive test sequences are formed by chaining the exact same atomic transitions used for training. Consequently, the test set explores the same state space as the training set, with no novel viewpoints or objects introduced. Thus, any performance degradation at $N > 1$ is attributable to a deficit in combinatorial generalization, rather than Out-of-Distribution (OOD) robustness.

## 5.2. Architectures

Three representative pre-trained backbones are evaluated to distinguish intrinsic architectural limits from objective-induced biases. **ViT** (Base: 86M, 12 Layers; Large: 307M, 24 Layers; Huge: 632M, 32 Layers), represents the canonical supervised Transformer optimized for semantic discrimination. **DINOv2** (Base: 86M, 12 Layers; Large: 300M, 24 Layers; Giant: 1.1B, 40 Layers) serves as a geometry-aware baseline, testing whether self-supervision overcomes structural deficits. Finally, **ResNet** (R-50: 25M, 50 Layers; R-101: 45M, 101 Layers; R-152: 60M, 152 Layers) provides a convolutional reference to contrast global attention mechanisms with local inductive biases. All encoders operate with frozen weights to evaluate the expressivity of their native representations.

## 5.3. Methodology

To rigorously assess whether these backbones have internalized the algebraic structure, a **Recursive Linear Probing** strategy is implemented. Specifically, a set of independent linear transition modules $\{T_{\phi,i}\}$ is assigned, each acting as an approximation for a corresponding group generator (for instance, the 6 generators in our evaluation naturally dictate the deployment of 6 independent linear probes). These probes are trained on atomic transformations ($N = 1$). This forces the probes to map the immediate local topology of the latent manifold. During testing, the probes coordinate to predict subsequent states along trajectories ($N \in \{2, \ldots, 20\}$) via a recursive readout mechanism, essentially mimicking algebraic iterated multiplication.

**Rationale for Linearity and Recursion.** The probes $\{T_{\phi,i}\}$ are strictly constrained to be linear maps applied recursively. Linearity is dictated by Group Representation Theory, which posits that a valid homomorphic embedding implies the existence of a linear representation $\rho(g)$. By allocating an independent linear map to each generator, the evaluation precisely models the discrete composition of physical transformations. This constraint ensures that successful reasoning is exclusively driven by the frozen backbone's intrinsic geometric structure. Crucially, the recursive execution—applying the atomic transitions step-by-step during inference—forces the evaluation to resolve the sequential dependencies causally, thus preventing computational shortcuts. Consequently, success across long sequence evaluations validates that the frozen backbone preserves the exact algebraic trajectory, rather than relying on shallow interpolation.

**Mechanism Verification via OPA.** To further validate the probe's findings and rule out optimization noise, OPA is employed. As a deterministic method, OPA acts as a direct check on intrinsic structural linearity, ruling out failures caused by training dynamics. A low or negative score con-clusively proves the absence of a valid linear operator for the transformation in the latent space, thereby attributing recursive collapse to architectural deficits rather than optimization issues.

## 5.4. Results and Analysis

Structural fidelity is evaluated by tracking the divergence between the recursively predicted latent state and the ground truth. To enable fair cross-architecture comparisons across latent representations with vastly different numerical scales, we normalize the prediction Mean Squared Error (MSE) against a **Global Mean Baseline**, contextualizing the absolute error into a universal Ratio Loss ($1 - R^2$). This baseline represents the MSE of a trivial predictor outputting the global mean of the target features (i.e., the target variance at step $N$). Approaching 1.0 indicates regression to variance noise, signifying a loss of specific spatial tracking capability. (Detailed benchmark settings, hyperparameters, and full quantitative tables are provided in the Appendix.)

**Main Takeaway.** The experiments reveal a universal complexity barrier: regardless of architecture, depth, or training objective, all models exhibit a fundamental degradation in tracking long-range dependencies in non-solvable groups. While abelian transformations result in slow, linear error drift, non-solvable transformations trigger a rapid and significant degradation. Crucially, empirical results demonstrate that scaling network depth yields negligible benefits; the degradation trajectories are virtually identical within the same architecture family, confirming that constant-depth scaling cannot resolve deep sequential dependencies.

**1. The Hierarchy of Complexity.** Figure 3(a) illustrates the absolute prediction error as a function of sequence length $N$. A consistent hierarchy emerges across all backbone families: **Level 3 $\gg$ Level 2 $>$ Level 1**. As $N$ increases, the loss on Level 1 and level 2 tends to drift linearly, maintaining reasonable structural coherence. In contrast, Level 3 exhibits a steeper, monotonic increase. Quantitatively, the absolute error on Level 3 is approximately $5.73\times$ to $17.72\times$ higher than on Level 1 (e.g., $5.73\times$ for DINOv2-Base).

**2. Rate of Structural Degradation.** Figure 3(b) further analyzes the speed of degradation by tracking the Normalized Loss against the dynamic Global Mean Baseline (Ratio Loss). The trajectories for Level 3 exhibit a significantly faster approach towards the trivial threshold compared to Level 2 and Level 1. On average, the Ratio Loss for Level 3 accumulates at approximately $2.43\times$ the rate of Level 2 across sequence lengths. This accelerated degradation distinguishes the failure from the simple cumulative error drift seen in Abelian tasks; it signifies a severe structural breakdown of the homomorphic mapping at shallow logical depths, consistently preventing reliable long-horizon reasoning.

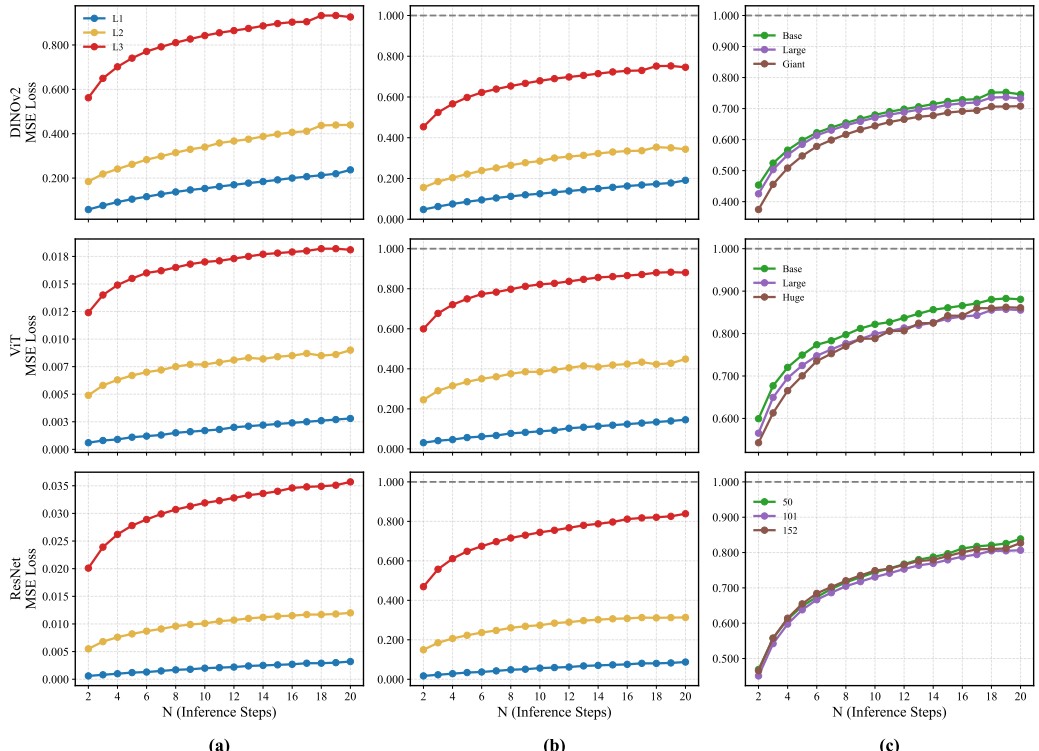

*Figure 3.* **Universal Structural Degradation on Non-Solvable Groups.** **(a)** Absolute prediction error vs. Sequence Length $N$: Non-solvable transformations consistently incur 5.73–17.72× higher error than Abelian ones. **(b)** Normalized loss relative to the Baseline: Level 3 trajectory approaches the baseline (dashed line) significantly faster than Level 1 and 2. **(c)** Normalized loss for Level 3 tasks across varying depths: Increasing model depth $L$ provides negligible performance gains.

**3. Insufficiency of Depth Scaling.** The impact of architectural capacity is analyzed in Figure 3(c), which tracks the Level 3 normalized loss across varying depths within the same model families. Results show that depth scaling exerts an almost imperceptible influence on the error trajectory, with curves for varying depths essentially overlapping. Increasing physical layers yields neither consistent nor significant reductions in Ratio Loss. This directly aligns with the theoretical prediction: stacking layers within a static computational graph ($\mathsf{TC}^0$) cannot facilitate the complexity class transition to $\mathsf{NC}^1$ required for modeling non-solvable groups.

**4. Deterministic Verification via OPA.** Finally, to rule out optimization failure, OPA is employed to deterministically test for linearity at the atomic level ($N = 1$). Results reveal a stark dichotomy: Abelian transformations (e.g., Translation X) yield strong linear fits (OPA $R^2 \approx 0.51$, Linear $R^2 \approx 0.79$), whereas non-solvable 3D rotations consistently fail (e.g., Rot Y: OPA $R^2 \approx -0.03$, Linear $R^2 \approx 0.32$). A negative $R^2$ implies that the optimal linear operator performs worse than a static mean predictor. This confirms that the failure is intrinsic: the latent space fundamentally lacks the geometry to represent non-solvable generators, making

subsequent recursive collapse inevitable.

## 6. Conclusion

This work confirms the intrinsic limits of Transformer image embeddings in non-solvable spatial reasoning. Theoretical analysis, bridging Group Theory and Circuit Complexity, establishes that constant-depth architectures lack the logical depth required to faithfully embed the algebraic structure of non-solvable groups. These complexity bounds are empirically validated via the Latent Space Algebra (LSA) benchmark. Through recursive linear probing, a significant structural degradation is observed in latent embeddings under non-solvable transformations, substantiating that the deficit is an intrinsic architectural property rather than a failure of optimization. Consequently, future research should reconcile the logical depth required for algebraic processing with the stability of neural training. Addressing this trade-off may require drawing inspiration from the biological principles of human navigation systems, solving the tension between computational expressivity and the recursive error accumulation inherent to deep sequential inference.

## Acknowledgements

We would like to thank Yalong Shi for the assistance with mathematical verification; Shujian Huang and Jie Guo for their constructive critiques regarding the argumentation structure and experimental design; Jingwei Xu for the valuable suggestions on experimental evaluation and improvements; and Yuanqi Li for the helpful feedback on writing and structural organization. Furthermore, we extend our sincere gratitude to the four reviewers for their meticulous review, encouraging appreciation of our topic, and insightful recommendations that significantly improved the experimental rigor and clarity of presentation. This work was supported by the Nanjing University Integrated Research Platform of the Ministry of Education, and the Fundamental and Interdisciplinary Disciplines Breakthrough Plan of the Ministry of Education of China.

## Impact Statement

This work presents a fundamental theoretical limitation of constant-depth Transformer architectures, demonstrating their inability to inherently model non-solvable spatial dynamics (e.g., 3D rotations) due to circuit complexity constraints.

**Safety and Reliability in Embodied AI.** As Vision Transformers are increasingly adopted as backbones for robotics and autonomous driving, our findings serve as a critical caution. We establish that standard ViTs, without recurrent depth or specific geometric inductive biases, effectively rely on shallow approximation rather than true algorithmic execution of spatial laws. In safety-critical applications—such as robotic manipulation or autonomous navigation—relying on such "pattern matching" representations could lead to unpredictable failures when the system encounters long-horizon compositional transformations not densely covered in training data. Our work suggests that, in these domains, incorporating hybrid modules or drawing inspiration from principles underlying biological navigation systems may be necessary to ensure logical robustness.

**Resource Efficiency and Environmental Impact.** By proving an intrinsic complexity barrier ($TC^0$ vs. $NC^1$), we highlight the futility of attempting to solve complex spatial reasoning solely via data scaling or parameter scaling within fixed-depth architectures. This theoretical insight has the potential to reduce the carbon footprint of AI research by steering the community away from resource-intensive brute-force training for tasks that are architecturally unsolvable, instead encouraging the design of more logically expressive and parameter-efficient models.

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

# A. LSA Benchmark Generation

To ensure reproducibility and rigorous evaluation of the Latent Space Algebra (LSA) benchmark, this section provides detailed specifications of the data generation pipeline and formalizes the mathematical isomorphism that guarantees structure preservation. We employ high-fidelity 3D meshes from the Stanford 3D Scanning Repository, specifically utilizing seven distinct object classes: Bunny, Dragon, Armadillo, Lucy, Happy Buddha, Asian Dragon, and Thai Statue. All observations are rendered as $224 \times 224$ grayscale images with fixed lighting and a zero-vector black background to eliminate environmental cues. To strictly control for operational complexity across all difficulty levels, the cardinality of the generator set is fixed to $|\Sigma| = 6$ for all experiments.

## A.1. Visual Injectivity and Mathematical Isomorphism

A standard 3D-to-2D projection ($\mathbb{R}^3 \to \mathbb{R}^2$) does not naturally preserve such algebraic structures. To formalize how we overcome this, we clarify the definition of our rendering function $\mathcal{R}$. While a standard camera maps arbitrary points in space to a 2D plane, in our benchmark, we render a fixed 3D object. Consequently, the resulting 2D image is parameterized entirely by the applied 3D rotation. Therefore, we abstract this rendering process as a function $\mathcal{R} : \mathrm{SO}(3) \to \mathrm{Im}(\mathcal{R})$, where the domain is the rotation group $\mathrm{SO}(3)$, and $\mathrm{Im}(\mathcal{R})$ is the set of all valid 2D renderings produced under all possible 3D rotations.

For the underlying algebraic group structure to be preserved in the visual domain, $\mathcal{R}$ must be a bijection. Because $\mathrm{Im}(\mathcal{R})$ is explicitly generated by mapping the elements of $\mathrm{SO}(3)$ through $\mathcal{R}$, surjectivity is inherently satisfied. The critical mathematical requirement is injectivity. This is exactly where our Visual Injectivity design comes into play: by deliberately utilizing asymmetric Stanford models, and by explicitly verifying our generated dataset to confirm that no two distinct 3D rotations produce identical rendered images, we ensure $\mathcal{R}$ is strictly injective, making the entire mapping a strict bijection.

To see why this bijection preserves the full computational complexity of the group, we first define how a single visual transformation operates. Because $\mathcal{R}$ is a strict bijection, it allows us to formally define the group action $\star$ of $\mathrm{SO}(3)$ on the image set $\mathrm{Im}(\mathcal{R})$. For any image $I = \mathcal{R}(h)$ and any rotation $g \in \mathrm{SO}(3)$:

$$g \star I := \mathcal{R}(g \circ \mathcal{R}^{-1}(I)) \tag{4}$$

This equation illustrates the physical ground truth: to appropriately apply a visual transformation $g$ to an image $I$, the system must implicitly decode the image back to its exact 3D pose via $\mathcal{R}^{-1}$, compose it with $g$, and re-render it.

Now, consider the sequential transitions in our LSA benchmark. Suppose the object starts at an initial pose, giving us the image $I_0 = \mathcal{R}(h_0)$. It is then asked to apply a sequence of discrete rotations $g_1, g_2, \ldots, g_N \in \mathrm{SO}(3)$ step-by-step. Therefore, predicting the final image after $N$ transformations requires computing:

$$I_N = g_N \star (\ldots (g_2 \star (g_1 \star I_0)) \ldots) = \mathcal{R}((g_N \circ \cdots \circ g_2 \circ g_1) \circ \mathcal{R}^{-1}(I_0)) \tag{5}$$

By construction, this induced group action on the 2D image space is isomorphic to the physical dynamics of $\mathrm{SO}(3)$. Since the visual mapping $\mathcal{R}$ provides no algebraic shortcuts, a neural network operating on these 2D images must implicitly evaluate the underlying combinatorial product $(g_N \circ \cdots \circ g_2 \circ g_1)$. Consequently, the visual reasoning task rigorously inherits the $\mathsf{NC}^1$-complete hardness associated with the transformation group.

## A.2. The Solvability Hierarchy of Datasets

**Level 1: Abelian (2D Translation).** This level establishes a baseline commutative structure isomorphic to the discrete translation group $\mathbb{Z}^2$. The generator set $\Sigma_{\mathrm{L1}}$ consists of six atomic translation operators with a fixed step size of $\delta = 20$ pixels. The action space is formulated using four cardinal translations (Right, Left, Up, Down) and two diagonal translations (Down-Right, Up-Left) to maintain the requisite cardinality of six. Mathematically, for an object located at a given coordinate $\mathbf{p} \in \mathbb{R}^2$, an action $g \in \Sigma_{\mathrm{L1}}$ applies the affine shift $\mathbf{p}' = \mathbf{p} + \mathbf{v}_g$. Since vector addition is commutative, the sequences generated by these operators form an abelian group, which is theoretically solvable by constant-depth circuits. Boundary conditions are handled by allowing objects to partially clip the frame, which effectively tests the model's capacity for object permanence without compromising or altering the underlying linear group logic.

**Level 2: Solvable Non-Abelian (2D Rigid Motion).** To introduce non-commutativity while retaining algebraic solvability, we construct a dataset governed by the Special Euclidean group $\mathrm{SE}(2)$. Restricting the group to 2D rigid motions resolves

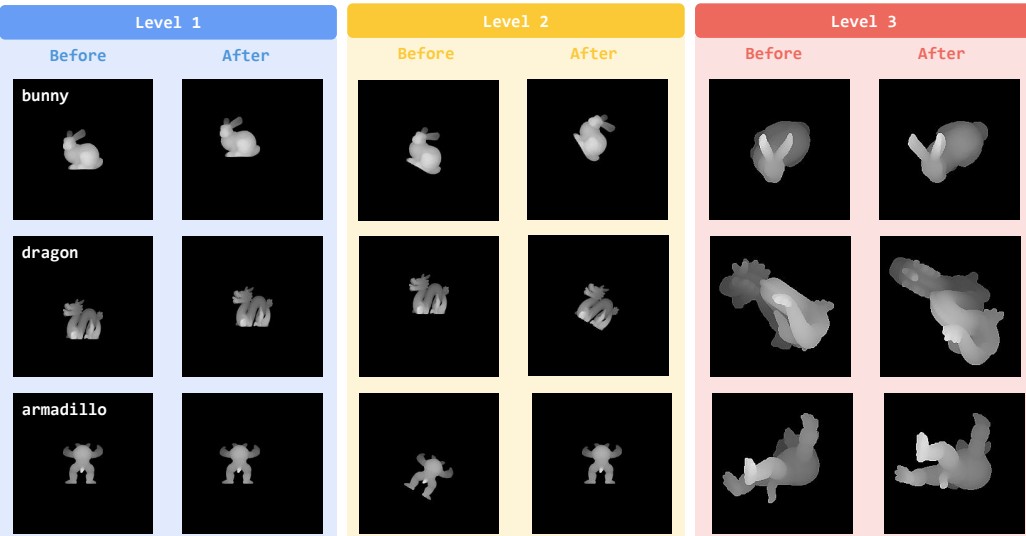

*Figure 4.* **Visualizing the Algebraic Hierarchy.** We display sample atomic transitions ($N = 1$) for the *Bunny*, *Dragon* and *Armadillo* objects across the three complexity levels. **Left (Level 1):** Pure 2D translations preserve orientation and scale. **Center (Level 2):** 2D Rigid Motions ($\mathrm{SE}(2)$) introduce in-plane rotations alongside translations, altering 2D planar orientation while strictly preserving the object's scale. **Right (Level 3):** 3D Rotations introduce out-of-plane transformations, revealing occluded geometry and fundamentally altering the visual topology, corresponding to the non-solvable $\mathrm{SO}(3)$ group structure.

this physical instability while precisely preserving the necessary algebraic conditions. The generator set $\Sigma_{\mathrm{L2}}$ comprises four translation operators identical to those in Level 1, supplemented by two in-plane rotation operators (e.g., clockwise and counter-clockwise rotations by a fixed angle around the image center). While rigid rotations and translations generally do not commute, the $\mathrm{SE}(2)$ group is structurally constrained such that its derived series terminates at the identity in finite steps. This progression evaluates the capability of architectures to process non-commutative operations that ultimately remain decomposable into finite abelian extensions.

**Level 3: Non-Solvable (3D Rotation).** This level directly targets the theoretical complexity boundary by embedding the non-solvable structure of the 3D rotation group $\mathrm{SO}(3)$. The generator set $\Sigma_{\mathrm{L3}}$ operates entirely in the 3D world space prior to the 2D rendering projection. We define six generators corresponding to extrinsic rotations around the orthogonal $X, Y$, and $Z$ axes by a fixed atomic angle of $\theta = \pm 30°$. Applying these generators iteratively produces a dense subset of the rotation group. Since this subset contains finite non-solvable subgroups isomorphic to the alternating group $\mathrm{A}_5$, it enforces an algebraic structure where operations are deeply entangled and fundamentally cannot be decoupled into independent abelian layers. Modeling long-horizon sequences in this non-solvable domain therefore presents an $\mathrm{NC}^1$-hard computational challenge, surpassing the theoretical expressivity limit of static, constant-depth execution graphs.

### A.3. Trajectory Sampling Strategy

To ensure robust and unbiased coverage of the group manifold, data generation employs a recursive random walk sampling strategy. For each object class, the object is initialized at a canonical pose, and a trajectory is constructed by sequentially applying random generators from $\Sigma$ up to a maximum depth of $N_{\max} = 20$. Every intermediate step of the trajectory is explicitly recorded, yielding a dataset of tuples $(I_0, S_{1:k}, I_k)$ for all $k \in \{1, \dots, 20\}$ that is comprehensive of state transitions for all intermediate sequence lengths.

To rigorously isolate combinatorial generalization from mere memorization, the models are trained exclusively on atomic transitions ($N = 1$) and are subsequently evaluated on the extended recursive sequences ($N \in \{2, \dots, 20\}$) extracted from these identical generative paths. This protocol guarantees that the test set explores the exact same state space as the training set without introducing novel viewpoints or out-of-distribution artifacts. Consequently, successful prediction during inference demands that the network architecture dynamically mimic algebraic iterated multiplication.

*Table 1.* Hyperparameters for Recursive Linear Probe.

| PARAMETER | TRAINING PHASE | TESTING PHASE |
|---|---|---|
| LEARNING RATE | $1 \times 10^{-4}$ | N/A |
| BATCH SIZE | 256 | 256 |
| EPOCHS | 100 | N/A |
| NUM WORKERS | 4 | 2 |
| SEED | 42 | 42 |

## B. Implementation Details

This section details the specific hyperparameters and training configurations employed for the Recursive Linear Probing experiments. All computations were executed using the PyTorch framework on NVIDIA GPUs.

### B.1. Backbone Specifications and Probe Architecture

Three representative backbones are evaluated to ensure diverse architectural coverage and to distinguish intrinsic architectural limits from objective-induced biases. **ViT** (Base: 86M, 12 Layers; Large: 307M, 24 Layers; Huge: 632M, 32 Layers) is selected to represent the canonical supervised Transformer optimized for semantic discrimination. **DINOv2** (Base: 86M, 12 Layers; Large: 300M, 24 Layers; Giant: 1.1B, 40 Layers) serves as a geometry-aware baseline, testing whether self-supervision overcomes structural deficits. Finally, **ResNet** (R-50: 25M, 50 Layers; R-101: 45M, 101 Layers; R-152: 60M, 152 Layers) provides a convolutional reference to contrast global attention mechanisms with local inductive biases. All encoders operate with frozen weights to evaluate the expressivity of their native representations.

To model the six algebraic generators independently, we employ a set of six separate Linear Probes, one dedicated to each generator. Each probe $T_\phi^{(i)}$ ($i = 1, \ldots, 6$) is a single linear layer that maps the concatenation of the image embedding $z$ to a predicted next-state embedding, where the projection maps from $2 \times d_{\text{model}}$ to $d_{\text{model}}$ (i.e., $T_\phi^{(k)} : \mathbb{R}^{2d_{\text{model}}} \to \mathbb{R}^{d_{\text{model}}}$). This per-generator design ensures that each probe specializes exclusively in one transformation, avoiding cross-generator interference and providing a cleaner measure of how well the representation encodes each individual algebraic transition.

### B.2. Training Configuration

A separate set of six probes is trained for each backbone variant. To ensure that performance differences reflect architectural properties rather than optimization noise, the random seed is fixed to 42 across all runs. The probes are optimized using the Adam optimizer with Mean Squared Error (MSE) loss, which minimizes the Euclidean distance between the predicted next-state embedding and the ground-truth target embedding, providing a direct measure of geometric divergence in the latent space. The specific hyperparameters for both training and recursive testing phases are listed in Table 1.

## C. Quantitative Analysis

To substantiate the claim regarding the non-solvable complexity barrier, we present the recursive prediction error normalized against the Global Mean Baseline. Specifically, we report the Ratio Loss $1 - R^2 = \mathcal{L}_{\text{model}}/\mathcal{L}_{\text{baseline}}$, where $\mathcal{L}_{\text{baseline}}$ is the MSE of a trivial predictor outputting the global mean of target features at step $N$ (i.e., the target variance). This normalization enables fair cross-architecture comparison by eliminating the effect of vastly different numerical scales across latent spaces. A value approaching 1.0 indicates that the model's predictions have regressed to the global mean, signifying a complete loss of structural tracking capability.

Table 2 reports the Ratio Loss for the Base variants of each architecture across all three complexity levels at steps $N = 5, 10, 20$. Two consistent patterns emerge. First, Level 3 (Non-Solvable) tasks incur substantially higher Ratio Loss than Level 1 (Abelian) or Level 2 (Solvable) tasks across all architectures, confirming that the non-solvable complexity barrier is not an artifact of absolute scale but a genuine structural property. Second, the Ratio Loss increases monotonically with $N$ for all models and complexity levels, indicating systematic error accumulation under recursive composition.

Table 3 further examines whether scaling model size alleviates the barrier, reporting Ratio Loss for Level 3 tasks across all architectural variants. Increasing model size yields marginal and inconsistent improvements: for instance, ViT-Huge achieves a slightly lower $N = 5$ ratio (0.70) compared to ViT-Base (0.75), yet the gap narrows and partially reverses at

$N = 20$ (0.86 vs. 0.88). DINOv2 exhibits the most consistent modest improvement with scale, while ResNet variants show negligible differences across depths. These results indicate that the non-solvable complexity barrier is largely invariant to model scale, pointing to an intrinsic limitation of supervised and self-supervised discriminative objectives rather than a capacity bottleneck.

*Table 2.* **Ratio Loss for Base Models Across Complexity Levels.** Values are normalized against the Global Mean Baseline at each step $N$, enabling fair cross-architecture comparison. A value of 1.0 corresponds to complete regression to the mean. Level 3 (Non-Solvable) consistently exhibits substantially higher Ratio Loss, confirming the algebraic complexity barrier.

| MODEL | COMPLEXITY LEVEL | $N = 5$ | $N = 10$ | $N = 20$ |
|---|---|---|---|---|
| VIT-BASE | LEVEL 1 (ABELIAN) | 0.0567 | 0.0877 | 0.1456 |
| | LEVEL 2 (SOLVABLE) | 0.3353 | 0.3850 | 0.4486 |
| | LEVEL 3 (NON-SOLVABLE) | 0.7494 | 0.8218 | 0.8807 |
| DINOV2-BASE | LEVEL 1 (ABELIAN) | 0.0858 | 0.1250 | 0.1910 |
| | LEVEL 2 (SOLVABLE) | 0.2214 | 0.2855 | 0.3435 |
| | LEVEL 3 (NON-SOLVABLE) | 0.5976 | 0.6796 | 0.7458 |
| RESNET-50 | LEVEL 1 (ABELIAN) | 0.0341 | 0.0566 | 0.0874 |
| | LEVEL 2 (SOLVABLE) | 0.2229 | 0.2737 | 0.3133 |
| | LEVEL 3 (NON-SOLVABLE) | 0.6484 | 0.7444 | 0.8386 |

*Table 3.* **Effect of Model Scale on Ratio Loss for Level 3 (Non-Solvable) Tasks.** Increasing model size yields only marginal improvements, demonstrating that the non-solvable complexity barrier is not resolved by additional capacity.

| ARCHITECTURE | SIZE | $N = 5$ | $N = 10$ | $N = 20$ |
|---|---|---|---|---|
| VIT | BASE | 0.7494 | 0.8218 | 0.8807 |
| | LARGE | 0.7247 | 0.7992 | 0.8553 |
| | HUGE | 0.7003 | 0.7882 | 0.8608 |
| DINOV2 | BASE | 0.5976 | 0.6796 | 0.7458 |
| | LARGE | 0.5847 | 0.6708 | 0.7326 |
| | GIANT | 0.5474 | 0.6443 | 0.7079 |
| RESNET | R-50 | 0.6484 | 0.7444 | 0.8386 |
| | R-101 | 0.6379 | 0.7307 | 0.8065 |
| | R-152 | 0.6548 | 0.7489 | 0.8269 |

## D. Detailed Analysis of Atomic Linearity (OPA)

To rigorously investigate the intrinsic algebraic structure of the learned representations, we employ Orthogonal Procrustes Analysis (OPA) alongside standard Linear Probing. This section details the experimental protocol and presents the comprehensive quantitative results for atomic transformations ($N = 1$).

### D.1. Experimental Protocol

Our analysis pipeline calculates whether a fixed geometric transformation $g$ can be approximated by a linear operator in the frozen latent space of the DINOv2-base backbone. The procedure consists of three cohesive stages.

**1. Data Sampling.** For a target object class (e.g., *Bunny*), we generate pairs of observations $(I_{\text{src}}, I_{\text{tgt}})$ linked by a specific transformation $g$. For **Level 1 (Translation)**, we sample random positions $(x, y)$ on a $224 \times 224$ canvas. Given a translation parameter $(\Delta x, \Delta y)$, the target image is rendered at $(x + \Delta x, y + \Delta y)$. In contrast, for **Level 3 (3D Rotation)**, we sample a random initial pose $P_a$ represented by a random rotation matrix from SO(3). The target pose is computed as $P_b = P_a \cdot R_g^T$, where $R_g$ is the fixed relative rotation matrix for the operator $g$ (e.g., 30° around Y-axis).

**2. Feature Extraction.** We utilize the CLS token output from the frozen DINOv2 encoder as the representation vector $z \in \mathbb{R}^{768}$. For each transformation task, we generate a dataset of $N = 3000$ pairs.

**3. Metric Definitions.** We evaluate the structural fidelity using two complementary metrics, employing an 80/20 train-test split for learned parameters. First, to assess **General Linearity**, we fit a standard Linear Regression model ($z_{\text{tgt}} \approx W z_{\text{src}} + b$)

and evaluate the coefficient of determination ($R^2$) on the test set. This metric determines if the transition allows for general affine distortions such as scaling or shearing. Second, to test for **Strict Isometry**—a requirement for structure-preserving group operations in a well-regularized space—we solve the Orthogonal Procrustes problem. We first center the data ($Z^c = Z - \bar{Z}$) and find the optimal orthogonal matrix $Q$:

$$Q = \operatorname*{argmin}_{\Omega:\Omega^T\Omega=I} \|Z^c_{\text{tgt}} - Z^c_{\text{src}}\Omega\|_F \tag{6}$$

The metric is reported as the standard $R^2$ score using consistent notation:

$$R^2_{\text{OPA}} = 1 - \frac{\sum \|Z_{\text{tgt}} - \hat{Z}_{\text{tgt}}\|^2}{\sum \|Z_{\text{tgt}} - \bar{Z}_{\text{tgt}}\|^2} \tag{7}$$

where $\hat{Z}_{\text{tgt}} = Z_{\text{src}}Q + \bar{Z}_{\text{tgt}}$. Crucially, a negative score indicates that the optimal linear rotation fits the data worse than the static mean baseline, serving as strong evidence against the existence of a linear operator.

### D.2. Full Quantitative Results

Table 4 presents the complete evaluation results. We observe a stark contrast between Abelian transformations (Translation, In-plane Rotation) and Non-Solvable transformations (Out-of-plane Rotation).

*Table 4.* **Atomic Linearity Analysis on DINOv2.** Comparison of general linear fit versus strict orthogonal fit. Note that while in-plane rotations (Rot Z) maintain positive OPA scores, out-of-plane rotations (Rot X, Y) consistently yield negative OPA $R^2$, confirming the absence of a valid linear operator.

| TRANSFORMATION | PARAMETER | LINEAR PROBE ($R^2$) | OPA ($R^2$) |
|---|---|---|---|
| *Level 1: Abelian (Translation)* | | | |
| TRANSLATION | X = 10PX | 0.7850 | 0.5084 |
| | X = 30PX | 0.7780 | 0.3432 |
| | X = 50PX | 0.7973 | 0.5021 |
| | XY = 30PX | 0.7074 | -0.0601 |
| *Level 3: 3D Rotation (Subgroup Analysis)* | | | |
| ROT Z (IN-PLANE) | 30° | 0.4495 | 0.3054 |
| | 60° | 0.4274 | 0.1708 |
| | 90° | 0.4480 | 0.1963 |
| ROT Y (OUT-OF-PLANE) | 30° | 0.3163 | -0.0286 |
| | 60° | 0.2782 | -0.2331 |
| | 90° | 0.2698 | -0.2526 |
| ROT X (OUT-OF-PLANE) | 30° | 0.3581 | 0.0168 |
| | 60° | 0.3259 | -0.1263 |
| | 90° | 0.3475 | -0.0395 |

