# OpenReview forum: "On the Intrinsic Limits of Transformer Image Embeddings in Non-Solvable Spatial Reasoning"
_ICML.cc/2026/Conference — ICML 2026 regular_

### Official Review · Reviewer_QvNH · 2026-03-03

**Soundness:** 2
**Presentation:** 3
**Significance:** 3
**Originality:** 3
**Overall Recommendation:** 4
**Confidence:** 2

**Summary:**

This paper theoretically analyzes the theoretical complexity required for functions to be able to capture non-solvable group structures. More specifically, the authors show that constant depth ViTs are unable to represent non-solvable subgroups of SO(3) and, therefore, are unable to represent SO(3) itself. To validate this theoretical result, the Latent Space Algebra benchmark is introduced, where 3 levels of increasingly difficult transformations are applied to the Stanford 3D scanning repository meshes, which are then projected to images. A model can be evaluated on this benchmark by testing whether its latent space is equivariant. This is tested by seeing if the latent representation of the base image can be transformed to the latent representation of the transformed image through a sequence of learned linear operations, given the sequence of operations applied to transform the image. The authors find that models indeed tend to struggle with capturing non-solvable group structure.

**Compliance With Llm Reviewing Policy:**

Affirmed.

**Final Justification:**

Firstly, I would like to thank the authors for their latest response. It helped a lot with understanding the experimental setup. The authors provided a very strong rebuttal in my opinion, that certainly addressed my concerns and that, when properly included in the paper, will significantly strengthen it. Therefore, I have increased my score.

**Key Questions For Authors:**

1. What is the (group?) structure in the image space for level 3 of LSA and how does it relate to the theory?
2. Why do the models perform so poorly on level 2 of LSA? I have seen the provided explanation in 5.4.2, but it is still not quite clear to me.
3. What is the link between lengths of words in SO(3) and depths/precision of ViTs?

**Limitations:**

Apart from what I have mentioned under weaknesses, yes.

**Strengths And Weaknesses:**

**Strengths**
- The paper is well-written. Despite my lacking background in complexity theory, the paper is written such that the idea and its motivation are quite clear.
- The theoretical part seems sound with clear proofs and and a properly structured story to combine all the theory.
- I think the theory provides an interesting perspective on the limits of modern deep learning architectures and the flaws of current approaches, as also mentioned by the authors in their concluding remark.

**Weaknesses**
- My main concerns lie in the LSA benchmark and the results obtained on it:
  * The third level of the benchmark consists of SO(3) transformations that are applied to the 3D meshes prior to projection. However, as the input to the vision models are simply images, these do not seem to constitute an actual group. To be specific, SO(3) forms a group on $\mathbb{R}^3$, but I believe it does not form a group on the projections of $\mathbb{R}^3$ onto some hyperplane, which is essentially what the input images are. Therefore, it is unclear to me how the theory provided in the paper relates to this specific case.
  * According to the theory, I would assume that solvable groups should not be a problem for ViTs. However, Figure 3b seems to indicate that the models perform worse on level 2 (solvable non-Abelian) than on level 3 (non-solvable). It is not clear to me why. If ViTs already fail to capture the structure of a group that they theoretically should be capable of capturing, then I think these experimental results cannot be used to empirically validate the given theory.
- It is not completely clear to me with respect to what a network should be logarithmically deep in order for it to capture the structure of SO(3). Since the paper focuses on vision models, that take single images as their inputs, should depth be increased with increasing resolution? I am similarly confused regarding the precision. As far as I know, precision is typically fixed during inference, so what is meant with polynomial precision?

If my concerns regarding the LSA benchmark are properly addressed, then I would of course be willing the improve my recommendation.

---

> ### Author Rebuttal · Authors · 2026-03-27
>
> We thank you for recognizing our theoretical soundness and originality. Your precise questions help clarify the mapping between abstract complexity and vision models. We address your points below, referencing our **General Responses (GR)**.
>
> **1. 2D Projections and $\mathrm{SO}(3)$ Structure.** Your concern is well-founded: $\mathrm{SO}(3)$ is not closed on 2D pixel space due to occlusion and projection. However, the group structure resides on the image set $\mathrm{Im}(\mathcal{R})$, not pixel space. By rendering strictly asymmetric objects (Visual Injectivity), $\mathcal{R}: \mathrm{SO}(3) \to \mathrm{Im}(\mathcal{R})$ is injective by construction and surjective by definition, hence a bijection—inducing a group action $g \star \mathcal{R}(h) := \mathcal{R}(g \circ h)$ isomorphic to $\mathrm{SO}(3)$. Since Lemma 4.3 reduces solely to whether the encoder can identify the underlying group element from $\mathcal{R}(g_N \cdots g_1)$, the full hardness of the Word Problem is preserved. We will formalize this in the revised appendix.
>
> **2. L2 vs. L3 Anomaly.** Your concern is entirely valid. As detailed in **GR4 (located below)**, this inversion was an artifact of non-isometric scaling. Our new controlled experiments eliminate this confounding factor: L3 now degrades significantly faster than L2, perfectly aligning with our theoretical hierarchy.
>
> **3. Word Problem, Depth & Precision.** Please see **GR1 (in Reviewer i2WB's section)** for the physical intuition linking the Word Problem and spatial computation: $N$ is the operation sequence length, not image resolution. **GR3 (in Reviewer cBjY's section)** details our proof, domain relations, and formal depth definition. Regarding precision, your intuition is correct. It denotes the bit-width of intermediate outputs, formalizing limits in boolean circuits to prevent "infinite-precision" loopholes. Logarithmic precision lets bit-width grow
> logarithmically with $N$. In practice, architectures use constant precision (e.g., FP32), perfectly fitting our theoretical bounds.
>
> **General Response 4: Resolving the Hierarchy Inversion (Level 2 vs. Level 3)**
>
> We sincerely thank the reviewers for their meticulous analysis regarding the degradation rate anomaly (L2 initially degrading faster than L3). Prompted by your critiques, we conducted a rigorous three-step ablation. The results conclusively prove this inversion was an artifact of data scaling drift, probe capacity limits, and baseline definitions. Due to space constraints, we summarize the methodology and conclusions below. Please refer to our updated figures for the full visual proofs: https://anonymous.4open.science/r/rebuttal-full-figure3steps-4C1B/README.md.
>
> * **Step 1: Eliminating Data Drift (Isometric Control).** We hypothesized that non-isometric scaling operators in our original L2 dataset ($\mathrm{Aff}(2)$) caused exponential drift, destabilizing the linear mapping. Replacing L2 with strictly isometric $\mathrm{SE}(2)$ successfully restored the correct hierarchy (L3 > L2). For a strictly fair comparison, we also stripped the auxiliary scaling operators from L3 (yielding a pure $\mathrm{SE}(2)$ vs. pure $\mathrm{SO}(3)$ evaluation). While the $\mathrm{L3} > \mathrm{L2}$ hierarchy held, the performance gap remained relatively narrow, leading us to investigate probe saturation.
> * **Step 2: Addressing Probe Capacity Bottleneck.** Following **Reviewer cBjY's** insightful suggestion, we recognized that a single shared linear probe lacks the bandwidth to optimally encode six distinct geometric operators. We deployed 6 independent, action-specific probes across both the original (with scaling) and the new pure isometric datasets. In all settings, the L3 degradation rate became faster than L2. This confirms that probe bandwidth constraints heavily masked the true structural gap.
> * **Step 3: Upgrading the Evaluation Baseline.** Finally, inspired by **Reviewer uHYw**, we refined our baseline metric to better reflect MSE optimization behavior. Our previous baseline evaluated the error relative to an identity mapping (predicting the initial state $I_0$). We upgraded this to the **Global Mean Predictor Baseline** (predicting the global spatial mean $\mu$), which mathematically corresponds to the $1 - R^2$ evaluation and strictly defines the zero-information collapse threshold. Under this standard, the rate at which L3 approaches the ceiling becomes starkly pronounced compared to L2.
>
> **Conclusion:** By systematically eliminating these artificial confounders, these ablations perfectly align our empirical observations with our theoretical $\mathsf{TC}^0$ complexity limits. The structural expressivity bottleneck for non-solvable groups is unequivocally confirmed. We will integrate this comprehensive analysis into the revision.

---

> > ### Author Rebuttal · Reviewer_QvNH · 2026-04-02
> >
> > Thank you for the additional experiments and clarifications. Particularly the new figures are interesting to see and have addressed most of my concerns. However, I'm not sure I fully understand the response regarding the applicability of the theory to images that are the result of a projection of a 3D mesh. For example, it is not quite clear to me what exactly $\mathcal{R}$ is here. Can you expand upon the given explanation a bit further?

---

> > > ### Author Response · Authors · 2026-04-03
> > >
> > > We sincerely thank you for the constructive attitude and continued engagement. The core question we address is exactly as you phrased it: how does the theory apply to "images that are the results of a projection of a 3D mesh"? Specifically, do these test images still possess the necessary group structure? To resolve this, we demonstrate that a rigorous group structure exists *within* this specific 2D image set space, and that this structure is strictly isomorphic to the external 3D group, making the task complexity mathematically equivalent.
> > >
> > > A standard 3D-to-2D projection ($\mathbb{R}^3 \to \mathbb{R}^2$) does not naturally preserve such algebraic structures. To formalize how we overcome this, we clarify the definition of our rendering function $\mathcal{R}$. While a standard camera maps points in space to a 2D plane, in our benchmark, we render a *fixed* 3D object. Consequently, the resulting 2D image is parameterized entirely by the applied 3D rotation. Therefore, we abstract this rendering process as a function $\mathcal{R}: SO(3) \to \mathrm{Im}(\mathcal{R})$, where the domain is the rotation group $SO(3)$, and $\mathrm{Im}(\mathcal{R})$ is the set of all valid 2D renderings produced under all possible 3D rotations.
> > >
> > > For the group structure to be preserved, $\mathcal{R}$ must be a bijection. Because $\mathrm{Im}(\mathcal{R})$ is explicitly generated by $\mathcal{R}$, it is inherently surjective. The critical requirement is injectivity. This is exactly where our **Visual Injectivity** comes into play: by deliberately utilizing asymmetric Stanford models, and by explicitly verifying our generated dataset to confirm that no two distinct 3D rotations produce identical rendered images, we ensure $\mathcal{R}$ is strictly injective, making the entire mapping a bijection.
> > >
> > > To see why this bijection preserves the full computational complexity of the group, we first define how a single visual transformation operates. Because $\mathcal{R}$ is a strict bijection, it allows us to formally define the group action $\star$ of $SO(3)$ on the image set $\mathrm{Im}(\mathcal{R})$. For any image $I = \mathcal{R}(h)$ and any rotation $g \in SO(3)$:
> > > $$g \star I := \mathcal{R}(g \circ \mathcal{R}^{-1}(I))$$
> > > This equation illustrates the physical ground truth: to apply a visual transformation $g$ to an image $I$, one must implicitly decode the image back to its exact 3D pose via $\mathcal{R}^{-1}$, compose it with $g$, and re-render it.
> > >
> > > Now, consider the sequential transitions in our LSA benchmark. Suppose the object starts at an initial pose, giving us the image $I_0 = \mathcal{R}(h_0)$. It is then asked to apply a sequence of discrete rotations $g_1, g_2, \dots, g_N \in SO(3)$ step-by-step. Therefore, predicting the final image after $N$ transformations requires computing:
> > > $$I_N = g_N \star ( \dots (g_2 \star (g_1 \star I_0)) \dots ) = \mathcal{R}( (g_N \circ \dots \circ g_2 \circ g_1) \circ \mathcal{R}^{-1}(I_0) )$$
> > >
> > > By construction, this induced group action is isomorphic to the physical dynamics of $SO(3)$. Since the visual mapping provides no algebraic shortcuts, a neural network operating on these 2D images must implicitly evaluate the underlying combinatorial product $(g_N \circ \dots \circ g_2 \circ g_1)$. Consequently, the visual reasoning task inherits the $NC^1$-complete hardness.

---

### Official Review · Reviewer_i2WB · 2026-03-09

**Soundness:** 3
**Presentation:** 4
**Significance:** 3
**Originality:** 4
**Overall Recommendation:** 5
**Confidence:** 4

**Summary:**

This paper argues that standard constant-depth Vision Transformers (ViTs) are intrinsically incapable of modeling non-solvable spatial transformations, such as 3D-rotations, due to fundamental circuit complexity limitations. The paper reduces the group homomorphism problem to the Word problem and shows that by Barrington's Theorem it is $NC^{1}$-complete, while constant-depth ViT with polynomial precision are known to lie within $TC^{0} \subsetneq NC^{1}$. To test it empirically, the paper designed the Latent Space Algebra (LSA) benchmark that tests model performance of the group homomorphism problem at three complexity levels - abelian, solvable non-abelian, and non-solvable groups. Empirical results validate the theorem.

**Compliance With Llm Reviewing Policy:**

Affirmed.

**Key Questions For Authors:**

1. How does the Non-Solvable Barrier theorem apply to approximate homomorphisms?
2. What could be an interpretation for Figure 3(b), where model performances at L2 continuously regrade pass the baseline for DINOv2 and ResNet? For ViT-base, is the similar behavior observed?

**Limitations:**

Yes

**Strengths And Weaknesses:**

Strength:

The paper explore the connection between group-theoretic solvability and neural circuit complexity classes and reveals a fundamental expressivity problem within the ViT architecture when it comes to group actions (particularly 3D rotation). The finding is significant in the context of visual reasoning and VLM models that employs ViT; it provides a theoretical structural hypothesis for spatial reasoning failures, which is further strengthen by discussing the limitation of some extensions to ViT that might have addressed the problem (such as chain-of-thoughts).

Applying circuit complexity characterization of the transformer to geomotric group modeling is novel; approaching the spatial reasoning task by bridging the algebraic and architectural complexity via the word problem is unique.

The paper is well written, the propositions and theorems are sound, and the empirical validation methods (including the synthetic data design) align with the theoretical claims.


Weakness:

The empirical validation employs pre-trained backbones and linear probe for validation. While the results indicates that all of pre-trained models performed poorly, it is not sufficient to isolate the cause to architectural expressivity. Since the assessment is recursive linear probe, representational linearity or other training-induced factors may also play a role. These confounding factors limit the validity of the empirical justification.

---

> ### Author Rebuttal · Authors · 2026-03-27
>
> We thank you for recognizing the novelty of bridging geometric group modeling with circuit complexity. Your insightful questions have profoundly improved our manuscript! We address your points below, referencing our **General Responses (GR)** for shared topics.
>
> **1. Pre-trained Architectures & Probes.** While fine-tuning models specifically for spatial reasoning is an ideal future direction, our findings remain robust. As detailed in **GR4 (in Reviewer QvNH's section)**, the expressivity gap holds perfectly even with action-specific independent probes. Furthermore, recalling our conclusion in Section 5.4: Orthogonal Procrustes Analysis (OPA) provides a deterministic complement at $N=1$ to rule out optimization failure. Abelian transformations yield strong fits (OPA $R^2 \approx 0.51$, Linear $R^2 \approx 0.79$), while non-solvable 3D rotations consistently fail (e.g., Rot Y: OPA $R^2 \approx -0.03$).
>
> **2. Approximate Homomorphisms.** Crucially, approximate homomorphisms do not alter the reducibility structure of our proof. The Word Problem reduction holds regardless of whether the target mapping is exact or approximate, since Barrington's Theorem operates on the structural existence of a faithful representation of the computation, not on its numerical precision. For why approximation fails to resolve the combinatorial bottleneck in practice, please see **GR2 (located below)**.
>
> **3. L2 vs. L3 Anomaly (Fig 3b).** Your observation was spot on. As detailed in **GR4**, this inversion was an artifact of non-isometric scaling. Our new controlled experiments eliminate this confounding factor: L3 now degrades significantly faster than L2, perfectly aligning with our theoretical hierarchy. *(Updated figure:https://anonymous.4open.science/r/rebuttal-full-figure3steps-4C1B/README.md)*
>
> **General Response 1: Clarifying the $\mathsf{TC^0}$ vs. $\mathsf{NC^1}$ Barrier and the Core Conjecture**
>
> We clarify the physical intuition behind our complexity framework and address our reliance on the $\mathsf{TC^0} \subsetneq \mathsf{NC^1}$ conjecture, which we will explicitly state as a conditional boundary in our revised manuscript.
>
> * **The Core Tension: Parallel vs. Sequential Capabilities.** The circuit complexity classes $\mathsf{TC^0}$ and $\mathsf{NC^1}$ fundamentally formalize our physical intuition regarding parallel versus sequential computing. Commutative spatial operations (e.g., 2D translations in abelian groups) can be aggregated simultaneously in parallel ($\mathsf{TC^0}$). However, certain spatial groups (e.g., non-solvable groups governing 3D rotations, such as $\mathrm{SO}(3)$) place strict demands on sequential computation: each transformation critically depends on the exact coordinate frame established by the entire history of prior steps. The sequential computational complexity of these spatial groups is therefore isomorphic to the $\mathsf{TC^0}$ vs. $\mathsf{NC^1}$ barrier.
>
> * **The Validity of the $\mathsf{TC^0} \subsetneq \mathsf{NC^1}$ Conjecture.** Relying on this conjecture is a necessary foundational anchor. If $\mathsf{TC^0} = \mathsf{NC^1}$, shallow parallel networks could simulate deep sequential logic without exponential size blow-up, collapsing the boundary between feed-forward parallel processing and recurrent architectures. While previous works have successfully relied on this boundary to characterize Transformer limits in formal logic—most notably in tasks like bracket matching (which has been empirically proven)—our core contribution is introducing this rigorous complexity barrier into the visual and physical domain.
>
> **General Response 2: The Necessity of Strict Group Homomorphism (Definition 4.1)**
>
> We thank the reviewers for their critical examination of Definition 4.1. Our primary motivation is to ground "spatial cognition" in rigorous mathematics, moving beyond phenomenological curve-fitting (e.g., learning a static mapping $f: \mathcal{X} \to \mathbb{R}^3$).
>
> * **The Algebraic Nature of Space.** Following Henri Poincaré's philosophy and Erlangen program, "space" is intrinsically defined by the group of continuous transformations acting upon it, not by static coordinates. While approximations suffice for narrow, bounded tasks (which we will clarify in the revision), evaluating true spatial cognition requires architectures to preserve these invariant group structures.
>
> * **Combinatorial and Sequential Bottlenecks.** The critical failure mode of approximate representations is combinatorial generalization. Without algebraic guarantees, networks merely memorize finite samples of the group manifold rather than internalizing its generative law, breaking down on novel compositional sequences. This failure amplifies catastrophically in sequential reasoning (e.g., world models or robotic navigation). A small heuristic deviation $\epsilon$ at each step compounds exponentially as $\epsilon_{\text{total}} \sim (1+\epsilon)^N$, causing severe divergence over a reasoning horizon $N$.

---

> > ### Author Rebuttal · Reviewer_i2WB · 2026-04-06
> >
> > I thank the authors for their response. The authors have addressed all of my questions. I will maintain my score as accept.

---

### Official Review · Reviewer_cBjY · 2026-03-10

**Soundness:** 3
**Presentation:** 2
**Significance:** 2
**Originality:** 3
**Overall Recommendation:** 4
**Confidence:** 4

**Summary:**

The submission takes a strong position on a timely question: whether the
difficulty Vision Transformers face on spatial reasoning tasks reflects a lack
of scale, or a more basic computational limitation. The paper formalizes
"robust spatial understanding" through a homomorphic latent representation,
requiring an encoder E to satisfy E(g · I) = ρ(g)E(I) for a faithful linear
representation ρ of the underlying transformation group. The main theoretical
claim is that once the group contains a finite non-solvable subgroup,
constructing such a representation is at least as hard as the finite-group
Word Problem, which is NC¹-complete. Combined with the standard placement of
constant-depth transformers with polynomial precision in TC⁰, the paper
concludes that constant-depth ViT encoders cannot realize this kind of spatial
representation, assuming the usual separation TC⁰ ⊊ NC¹.

The empirical evaluation introduces the Latent Space Algebra benchmark, with three
progressively harder regimes: 2D translations, 2D affine transformations, and
3D rotations. Frozen pretrained representations from ViT, DINOv2, and ResNet
are evaluated using recursive linear probes trained on one-step transitions
and tested on longer compositions. The reported pattern is that performance
degrades much more sharply on the most complex transformation family, which
the paper interprets as evidence that current latent spaces do not preserve
the compositional structure needed for spatial reasoning.

**Compliance With Llm Reviewing Policy:**

Affirmed.

**Final Justification:**

The author has adequately addressed the issues I raised.

**Key Questions For Authors:**

1. What is the exact input to the computational problem in Lemma 4.3? If the
   encoder receives the rendered image I_N rather than the generator sequence
   (g₁, ..., g_N), please show the explicit reduction from the encoder task
   to the Word Problem. This is the critical gap in the current proof.

2. Why does a negative OPA R² imply the absence of a valid linear operator in
   GL(d, ℝ)? OPA is constrained to orthogonal maps, not general invertible
   ones. Please clarify what the OPA results establish relative to the
   theoretical object in Definition 4.1.

3. Does restricting Level 3 to pure rotational generators change the
   conclusions? A benchmark using only the three rotation axes would be a
   cleaner empirical test of the non-solvable claim.

**Limitations:**

No.

The paper should be more explicit about its limitations. In particular, it should acknowledge that the complexity-theoretic conclusion is conditional and presently depends on a reduction that is not fully spelled out, that Definition 4.1 is a strong modeling choice rather than an established necessity condition, and that the empirical results may reflect probe design and distribution shift as much as the intended algebraic effect.

**Strengths And Weaknesses:**

Strength:

1> Using group solvability as the lens for spatial composition is a genuinely fresh idea, and it gives the paper a clear conceptual identity that stands out in this literature.

2> The benchmark is designed with a point of view rather than being a generic task collection, which makes it considerably more interesting as an empirical contribution.

3> The question tackled by this paper is timely and the stakes are real — if the failure is architectural rather than a data problem, that changes how the community should be spending its effort.

Weakness:
1> The central theorem is not yet established at the precision the paper needs — the hard input in the Word Problem is a word over generators, but the encoder receives a rendered image, so the difficult compositional step may already have been carried out before the encoder is ever invoked.

2> The paper overstates the status of its own formalization — Definition 4.1 is a clean and appealing way to express structured spatial representation, but calling it a necessary condition for spatial cognition goes further than the argument supports.

3> The recursive probe uses a shared linear map with an action-dependent offset rather than a family of action-specific operators, which is a serious mismatch with the theoretical object — poor long-horizon behavior under this design may reflect a limitation of the probe, not the backbone.

4> Level 3 is not a pure test of the non-solvable claim since it mixes rotation generators with solvable scaling and translation, and training only on N=1 transitions from a canonical pose means longer test trajectories almost certainly introduce unseen viewpoints.

---

> ### Author Rebuttal · Authors · 2026-03-27
>
> Thank you for your rigorous review; your insights strengthened our paper. We address your concerns below, referencing our **General Responses (GR)**.
>
> **1. Theoretical Concerns**
>
> * **The Proof Gap & Implicit Info:** While the renderer easily computes physics, the core question is if the *encoder* intrinsically captures this algebraic structure. It could trivially map all images to one point. Since the encoder processes a single *final* image yet must embed it at the exact coordinate defined by the algebraic relation, a *single forward pass* must *implicitly* solve the Word Problem. See **GR3 (located below)** for the formal reduction and **GR1 (in Reviewer i2WB's section)** for physical intuition.
> * **OPA's Validity:** You are mathematically correct: A negative $R^2$ proves the absence of an isometric structure, not necessarily a skewed $\mathrm{GL}(d, \mathbb{R})$ representation. Thus, we pair OPA with **Learned Linear Probes** (testing $\mathrm{GL}(d, \mathbb{R})$). In **Appendix Table 4**, our data shows a dual failure: for complex rotations (e.g., Rot Y $90^\circ$), the unconstrained probe yields a weak $R^2 \approx 0.27$ (degraded $\mathrm{GL}(d, \mathbb{R})$ structure), while OPA collapses to $R^2 = -0.25$, proving the lack of an isometric core.
> * **Strictness of Definition 4.1:** While approximate heuristics work for bounded tasks, compounding errors inevitably cause combinatorial failures on long sequences. See **GR2 (in Reviewer i2WB's section)** for our spatial definition's justification.
>
> **2. Experimental Concerns**
>
> * **Probe Limits & Level 3 Purity:** Your suggestion regarding group purity was excellent. We re-ran the mentioned experiments; the updated results perfectly match our expectations and resolve the L2/L3 inversion. Please see **GR4 (in Reviewer QvNH's section)**.
> * **Unseen Viewpoints (OOD):** You noted that $N=1$ training might introduce unseen viewpoints. However, as detailed in Section 5.1, our data uses continuous random walks of length 20. Training on atomic transitions ($N=1$) extracted *from these exact trajectories* (e.g., step 18 to 19) ensures training and testing explore identical state spaces. With no unseen viewpoints, degradation reflects an algebraic combinatorial generalization failure.
>
> **General Response 3: Clarifying the Proof Reduction, "Constant Depth," and the Word Problem**
>
> We thank the reviewers for their rigorous examination. Misunderstandings of Theorem 4.5 and Lemma 4.3 largely stem from conflating a "single forward pass" and the "Word Problem." We clarify these crucial distinctions:
>
> **1. The Single Forward Pass Paradigm**
>
> Consider images connected by sequential group transformations (e.g., A rotates to B, then C). Mainstream encoders map the images to their respective latents $z_{\text{A}}, z_{\text{B}}, z_{\text{C}}$ via a single forward pass. A faithful spatial embedding requires these latents to satisfy certain algebraic constraints. Our framework investigates the intrinsic limit of this paradigm: can a single forward pass truly embed images into a space governed by such algebraic constraints?
>
> **2. Connection to the Word Problem (Lemma 4.3)**
>
> The computational hardness of non-solvable groups arises from strict sequential dependencies: non-commutative operations mean each state depends rigidly on all prior outcomes. The Word Problem formalizes this problem of sequential dependence, while Barrington's Theorem establishes its computational complexity, requiring a circuit depth that grows with $N$—**the transformation sequence length, strictly *not* the image resolution.**
>
> Why does processing a *single image* trigger this sequence-level complexity? As detailed in Lemma 4.3, the encoder processes the single final image $I_N$, not the sequence of transformations. To place this image at its correct algebraic coordinate $z_N$, the output must satisfy $z_N = \rho(g_N \dots g_1)z_0$. To map $I_N$ to this mathematical destination, the representation produced by the single forward pass must intrinsically contain all the algebraic information necessary to determine the iterated product of those $N$ transformations. Because successfully embedding this complete information over a finite group is computationally equivalent to solving the Word Problem, this capability strictly requires at least $\mathsf{NC^1}$ complexity. Since standard ViTs operate within $\mathsf{TC^0}$, and $\mathsf{TC^0} \subsetneq \mathsf{NC^1}$ (under standard conjectures), a single forward pass fundamentally cannot satisfy these constraints.
>
> **3. Clarifying $N$ and "Constant Depth"**
>
> In circuit complexity, "depth" is absolutely not a fixed number; rather, it describes the asymptotic scaling of the computational path relative to the input size $N$. A deployed ViT/ResNet has a completely static graph. Whether the physical sequence $N$ is 10 or 10,000, the network executes the exact same fixed operations. It cannot dynamically scale its internal depth to match the growing sequential demand.

---

> > ### Author Rebuttal · Reviewer_cBjY · 2026-04-03
> >
> > I thank the author for adequately addressing the issues I raised. I will raise the score.

---

### Official Review · Reviewer_uHYw · 2026-03-17

**Soundness:** 2
**Presentation:** 3
**Significance:** 3
**Originality:** 3
**Overall Recommendation:** 4
**Confidence:** 3

**Summary:**

This paper argues that ViTs' spatial reasoning failures in mental rotation are architectural and not data-driven. They formalize spatial understanding as learning a group homorphism. Then, they show that for non-solvable groups like SO(3), this requires NC^1 computation. However, constant depth ViTs are limtied to TC^0, creating a complexity barrier. They validate this with a new benchmark (lSA) to test three algebraic dfficulty levels, finding that all architectures (ViT, DINOv2, ResNet) maintain structure for abelian transformations but collapse on non-solvable ones, with depth scaling providing no remedy.

**Compliance With Llm Reviewing Policy:**

Affirmed.

**Key Questions For Authors:**

1. There's a proof gap-- Lemma 4.3 constructs a solver using N encoder calls plus matrix composition, which is not TC^0. How does this now work for a single forward pass?
2. Level 2 exceeds Level 3 failure for most models, inverting the solvabilty hierarchy. How do you explain this?
3. Under your framework, ResNet-152 has more than enough depth for NC^1. Why doesn't it overcome the barrier?
4. No model ever crosses failure threshold on Level 3, just degradation. What is the criterion for "structural collapse"?

**Limitations:**

- Data partially contradicts thesis. The solvability hierarchy inverts under normalized metrics; no model actually collapses on Level 3 by teh paper's own threshold. The OPA dichotomy seems to be actually in plane vs out of plane opposed to solvable vs non-solvable.
- Frozen encoders + linear probes != architectural capacity. The theory claims ViTs _can't_ learn solvable structure-- the experiments show that ViTs pretrained on ImageNet didn't learn it as tested by a single linear layer, which is a very different claim. No fine-tuning, no nonlinear probes, no positive controls.
- No constructive contribution-- identifies a problem but proposes no fix, tests no architecture, and offers to guidance
- The paper doesn't really discuss that approximate, task-specific, heuristic, or non-linear spatial representations could suffice in practice

**Strengths And Weaknesses:**

Soundness

Strengths:
- Overall this is a nice and elegant paper. The paper's three-solvability hierarchy (abelian -> solvable non-abelian -> non-solvable) is grounded in well-established group theory. The mapping from Level 1 to Leve 2 to Leve 4 is algebraically correct and a clean progression.
- The use of OPA (Orthogonal Procustes Analysis) as a deterministic, optimization-free verification of linearity in latent space is methodologically sound and a valuable complement to the learned probing appraoch. The negative R^2 values for out-of-plane rotations in Table 4 (Appendix) are are particularly convincing as well. They demonstrate the best orthogonal linear probe operator performs worse than a static mean predictor, ruling out the possibility that the probe training simply failed to find the right linear map.
- The experimental controls are carefully designed for visual injectivity, combinatorial generalization, which cleanly isolated algebraic structure-tracking from OOD robustness. It's a well thought out experimental design.
- The consistency of results across three architecture families, with multiple scales within each family, and both probing and OPA methods strengthens the empirical claims quite a lot.

Weaknesses:
- There appears to be a logical gap in the central proof (theorem 4.5). In Lemma 4.3, the encoder receives a generator sequence (g_1, ..., g_N) as explicit input. However, in actuality, the encoder processes a single image, not a generator sequence. Lemma 4.3, from my understanding, constructs a Word Problem solver by making N separate encoder calls, and then composing the resulting representations. The composite computatoin (N oracle calls to the TC^0 function plus matrix composition is not itself TC^0). The reduction therefore does not yield the contradiction of NC^1 ⊆ TC^0 as claimed. A TC^0 function called N times with external compositionhas greater power than TC^0 alone.
- The entire result is ocnditional on the unproven conjecture that TC^0 ⊊ TC^1. The paper is, to its credit, transparent about this but also this means the complexity barrier is not a theorem in the unconditional sense. This is fine but readers should understand that this is conjecture.
-  The solvability hierarchy breaks in a normalized metric. In Table 2, the DINOv2-Base Level 2 error exceeds its identity baseline giving a ratio of ~1.11, crossing failure threshold. But Level 3 error remains _below_ its baseline. The same inversion occurs for ResNet-50, where Lvel 2 fails but Level 3 doesn't fail. So for 2 out of the 3 architectures, the "solvable" level crosses the failure threshold while the "non-solvable" level does not. This directly contradicts the predicted hierarchy. The paper acknowledges this anomaly ("Level 2 sometimes degrades faster than Level 3...") but attributes it to "non-isometric scaling causing magnitude instability" without further investigation. This deserves substantially more attention than just that line.
- The structural collapse is also overstated. Table 3 shows every normalized loss for Level 3 is below 1.0, which, to my understanding, means that every model at every depth at every tested N still outperforms the identity basleine. No model ever  actual fails in the paper's own terms. We see a degradation, not a breakdown.
- It seems like the depth-scaling argument is internally inconsistent with the complexity framework. The paper argues that increasing depth (e.g. ResNet-50 -> ResNet-152) doesn't help because it doesn't change the complexity class. But "constant depth" in circuit complexity means O(1) w.r.t. input size n. For 224 x 224 images, log_2(n) is 16. ResNet-152 has 152 layers... which is more than logarithmic depth. Shouldn't the model then have sufficient depth to simulate NC^1 circuits? The fact that it doesn't help suggests the bottleneck is optimization or representation capacity, not the expressivity barrier like the paper claims.
- Definition 4.1 (Homomorphic Sptial Embedding) is assumed as necessary for spatial understanding, but the assumption is not justified. Models _can_ achieve practically useful spatial reasoning through approxmimate, heuristic, or task-specific representations that still aren't faithful group homomorphisms.
- There is a conflation between architecture and pretraining data (i.e. all tests are on pretrained representations, not what the architecture is capable of when optimized for spatial tasks)
- DINOv2-Base errors seem x30-40 larger than ViT-Base which may reflect different embedding norms rather than different spatial reasoning quality. The paper doesn't normalize for this which makes cross-architecture comparisons difficult to interpret

Presentation

Strength:
- The overall narrative arc is very compelling and well-structured logically.
- Figure 1 effectively communicates both the motivating problem (ViTs failing mental rotation) and the formal framework (right: homomorphism alignment question). Figure 2 clearly explaisn the recursive probing protoco.
- The three level solvability hierarchy is an intuitive and memorable organizing principle for both the theory and experiments.
- The anticipating counteraguments (chain-of-thought, positional encodings) shows really nice thought process + intellectual honesty

Weaknesses
- Key terms are introduced without citation or definition, which is quite confusing to someone who is not as familiar with this space. Barrington's Theorem, the Word Problem, and circuit complexity classes all appear in the abstract and introduction without any citatoins. For an ICML audience, they need immediate gorunding-- Many readers will not know what TC^0 or NC^1 are upon encountering them in the abstract.
- "Constant depth" is never explicitly defined for the ML audience-- it's a load bearing concept. Without a proper definition, the first instinct of many readers will be to conflate depth (# of layers) with circuit depth (asymptotic scaling).
- The LSA benchmark is not clearly introduced as a novel contribution on the first mention. THe paper should say "We introduce" or "we propose" on the first mention.

Significance
- Very important and timely question. As ViTs become backbones for robotics, autonomous driving, embodied AI, understanding whether spatial reasoning failures are data-fixable or achitecture-intrinsic is a first-order question
- the solvability hierarchy is a useful conceptual framework that could organize future research on spatial reasoning benchmarks. The abelian/solvable/non-solvable taxonomy gives the community a principled way to categorize spatial tasks by algebraic difficulty
- The LSA benchmark fills a gap and could even be emphasized as a contribution! Existing spatial reasoning benchmarks test downstream task performance but don't isolate algebraic structure in representations.
- The OPA negative-R^2 finding for out-of-plane rotatoins is a very independently useful result-- that standard ViT latent spaces lack isometric linear structure for 3D rotations

Weaknesses:
- No constructive guidance-- the paper identifies a barrier but proposes no solution. What architecture would overcome it?

Originality

Strengths:
- The central insight-- connecting group solvability to circuit complexity for understanding ViT spatial reasoning-- is novel. Prior work on ViT expressivity focuses on formal languages; prior work on spatial reasining failures is empirical. This paper bridges the two through group theory, which is a creative synthesis
- The three-level algebraic ttaxonomy is novel and useful organizational contribution going beyond ad-hoc task difficult categorization
- Using OPA as a deterministic complement to learned probing is to my knowledge novel when used in this way

Weaknesses:
- Paper does not adequately position itself against the group representation learning literature, the disentanglement literature, and the equivariant network literature-- which makes originality harder to assess here

---

> ### Author Rebuttal · Authors · 2026-03-28
>
> We thank you for recognizing the novelty of bridging geometric group modeling with circuit complexity. Your insightful questions and incredibly encouraging review have profoundly improved our manuscript. We address your points below, referencing our **General Responses (GR)** for shared topics.
>
> **1. Gratitude and Adoption of Suggestions.**
>
> We are deeply moved by your professional and warm review. Thank you for validating our core motivation: utilizing group algebra to elevate "spatial reasoning" from phenomenological descriptions to rigorous mathematical structures (inspired by Poincaré and Klein; see **GR2 (in Reviewer i2WB's section)**). We also highly appreciate your recognition of our experimental design, particularly the OPA method. Furthermore, we earnestly accept your presentation suggestions. To bridge the cross-disciplinary gap, our revised introduction will immediately define key terms and clarify potentially confusing complexity concepts (e.g., the input scale $N$ and "constant depth"). Finally, following your excellent advice, we will prominently highlight the LSA benchmark as a novel contribution.
>
> **2. Clarifying Theoretical Misunderstandings.**
>
> Regarding the theoretical mappings, we address your concerns across three specific points:
>
> * **Single Forward Pass vs. $N$ Encoder Calls.** You are entirely correct that making multiple calls to a $\mathsf{TC^0}$ circuit yields strictly greater power than $\mathsf{TC^0}$ alone. However, Lemma 4.3 models a single forward pass—the standard embedding paradigm—not $N$ iterative calls. The encoder never receives the generator sequence explicitly. Instead, our proof shows that if the encoder maps a single final image to the correct algebraic latent, it must implicitly contain all the information required to solve the Word Problem. This yields the desired contradiction. We strongly recommend reviewing **GR1 (in Reviewer i2WB's section)** for the physical intuition behind this, and **GR3 (in Reviewer cBjY's section)** for the formal reduction logic.
> * **The Definition of Depth and Input Scale $N$.** Your observation regarding ResNet-152 stems from a subtle conflation of the input scale $N$ (action sequence length) with image resolution. As you correctly noted, circuit depth refers to an asymptotic scaling relative to $N$. While ResNet-152 has many layers, its circuit is static and cannot dynamically grow with $N$; thus, its depth relative to $N$ remains strictly $\mathcal{O}(1)$. We kindly recommend consulting **GR3** for a detailed clarification of these formal domain relationships.
> * **Approximate Homomorphisms and the $\mathsf{TC^0} \subsetneq \mathsf{NC^1}$ Conjecture.** Regarding the necessity of strict homomorphisms (Definition 4.1) and the behavior of approximate representations, please see our response to **Reviewer i2WB** and the detailed discussion in **GR2**. Furthermore, our reliance on the standard but unproven $\mathsf{TC^0} \subsetneq \mathsf{NC^1}$ conjecture, along with its isomorphism to the computational tension of spatial reasoning, is explicitly addressed in **GR1**.
>
> **3. Clarifications on Experimental Anomalies and Baselines.**
>
> We appreciate your scrutiny, which inspired our rigorous re-evaluation.
>
> * **L2 vs. L3 Inversion:** Hypothesizing scaling drift was insufficient. As detailed in **GR4 (in Reviewer QvNH's section, Steps 1-2)**, our strict ablations—isolating purely isometric groups and deploying independent probes—resolved this anomaly. Stripped of these artifacts, L3 unequivocally degrades faster than L2, aligning with our theory.
> * **Baseline & "Collapse":** We concede "collapse" was overstated since original ratios never exceeded 1. To accurately measure expressivity, we adopted the **Global Mean Predictor Baseline** ($1-R^2$; see **GR4 Step 3**). While L3's ratio tightly approaches 1.0 (indicating severe convergence to the zero-information mean), this metric is statistically rigorous. Crucially, the pronounced L3 vs. L2 degradation gap establishes a robust standard for distinguishing encoded algebraic structures.
> * **Cross-Architecture Norms:** The 30-40x absolute MSE gap between DINOv2 and ViT reflects distinct scale norms. Our $1-R^2$ ratio naturally resolves this. By dividing absolute MSE by each model's global embedding variance, we factor out these scale artifacts. This scale-agnostic metric enables fair cross-architecture comparisons, revealing a remarkably consistent L3 degradation pattern universally.
>
> **4. Future Directions.**
>
> Rather than proposing quick patches, we offer a foundational diagnosis. Although $k$ iterative calls (e.g., Chain-of-Thought) could bypass the constant-depth bottleneck, they risk inheriting traditional RNN limitations. As a radical future direction, we envision moving beyond static feedforward models to explore internal continuous path integration, akin to human spatial navigation.

---

> > ### Author Rebuttal · Reviewer_uHYw · 2026-04-04
> >
> > Thank you for the detailed rebuttal. The concession on the "collapse" framing, the isometric-group ablation for the L2/L3 inversion, and the normalized cross-architecture metric are appreciated improvements. The single forward pass clarification is helpful-- I accept that the encoder processes one composed image rather than making N calls.
> >
> > However, two concerns remain unaddressed.
> >
> > First, the theory claims an architectural expressivity barrier, but the experiments test frozen pretrained encoders with linear probes. These measure what pretrained representations happen to contain, not what architecture can learn when optimized for the task. Finetuning on L3, nonlinear probes, or a positive control architecture would directly test this claim.
> >
> > Second, the paper still does not position itself against the equivariant network literature which is the most natural related work for a paper claiming standard architectures cannot learn gorup structure-- and the most obvious source of positive control baselines.
> >
> > I maintain my score at 4.

---

> > > ### Author Response · Authors · 2026-04-07
> > >
> > > We appreciate the reviewer’s continued engagement. Your feedback has been instrumental in refining our claims. Due to the tight rebuttal window, we prioritized the **non-linear probe experiment** to address your concern regarding architectural capacity. Our responses to your remaining questions are detailed below.
> > >
> > > **1. Architectural Capacity vs. Pretrained Features**
> > >
> > > We concede that evaluating frozen encoders with linear probes tests the capabilities in typical use cases, rather than the maximum representational potential of the architecture. To address this, we conducted new experiments using non-linear probes by introducing  hidden ReLU activation layers (for Figures, see: https://anonymous.4open.science/r/rebuttal-round2-nonlinearprobe-3F7E/README.md).
> > >
> > >  For ViT-based architectures (DINOv2, ViT-Base), the degradation on Level 3 is still faster than on Level 2, and the normalized ratio crosses the threshold ($Ratio > 1.0$). This indicates that even when the readout mechanism's capacity is enhanced, the bottleneck persists, aligning with our original expectations. Interestingly, for the ResNet architecture, Level 2 degrades faster than Level 3 after $N=16$ (absolute loss) and approaches the baseline faster after $N=14$ (ratio loss). We attribute this to the inductive bias of standard convolutions. CNNs operate on a fixed spatial grid. Since Level 2 evaluates the $\mathrm{SE(2)}$ group, the recursive application of in-plane rotations disrupts the local receptive field assumptions of the fixed grid. In contrast, ViTs process tokens globally without such grid constraints, allowing their degradation pattern to better reflect the underlying algebraic solvability hierarchy.
> > >
> > > We emphasize that our initial rationale for employing a linear probe was to strictly evaluate whether the latent space inherently formed a linear representation, thereby isolating the intrinsic geometric structure of representation from the probe's computational capacity. Furthermore, we acknowledge that our interpretation of the ResNet anomaly remains a hypothesis. The observed inversion might stem not solely from architectural differences, but also from potential representational shifts introduced by the ReLU probe itself, which warrants further rigorous empirical validation.
> > >
> > > **2. Equivariant Networks**
> > >
> > > Your suggestion to include Equivariant Networks as a positive control is well-taken. As discussed in Section 4.4.3, their geometric capabilities stem from analytical priors rather than standard optimization. We initially excluded them from the empirical baselines due to modality mismatches, as most $\mathrm{SE(3)}$ networks operate on 3D point clouds rather than 2D RGB images, making a direct comparison difficult.
> > >
> > > Finally, we will clarify the relevant limitations in the final version. In future work, we plan to expand this framework to test alternative architectures like State Space Models and continue to investigate the mechanisms underlying the formation of spatial representations.

---

### Decision · Program_Chairs · 2026-04-30

**Decision:**

Accept (regular)

**Comment:**

This paper analyzes failures of vision transformers at spatial reasoning tasks such as mental rotations. The framing of this paper is that these issues are by and large due to architectural limitations and not due to a lack of supporting training data (as is often assumed). by formalizing spatial understanding as learning a group homorphism they are able to prove that ViTs fall short. This resulted is strengthened empirically through a new benchmark that tests vision architectures at three algebraic difficulty levels.

This paper received several detailed reviews, and after the rebuttal most concerns are addressed. Reviewers highlight the framing and execution of the paper (described as nice, elegant, and fresh) as well as the clear contribution that is quite timely. Remaining concerns are largely minor. Overall I think this paper is a strong contribution.